# Event-Triggered Time-Varying Bayesian Optimization

**Paul Brunzema**                                                    *paul.brunzema@dsme.rwth-aachen.de*
*Institute for Data Science in Mechanical Engineering*
*RWTH Aachen University*

**Alexander von Rohr**                                                    *vonrohr@dsme.rwth-aachen.de*
*Institute for Data Science in Mechanical Engineering*
*RWTH Aachen University*

**Friedrich Solowjow**                                                    *friedrich.solowjow@dsme.rwth-aachen.de*
*Institute for Data Science in Mechanical Engineering*
*RWTH Aachen University*

**Sebastian Trimpe**                                                    *trimpe@dsme.rwth-aachen.de*
*Institute for Data Science in Mechanical Engineering*
*RWTH Aachen University*

**Reviewed on OpenReview:** *https://openreview.net/forum?id=WEYMCLu8u7*

## Abstract

We consider the problem of sequentially optimizing a time-varying objective function using time-varying Bayesian optimization (TVBO). Current approaches to TVBO require prior knowledge of a constant rate of change to cope with stale data arising from time variations. However, in practice, the rate of change is usually unknown. We propose an event-triggered algorithm, ET-GP-UCB, that treats the optimization problem as static until it detects changes in the objective function and then resets the dataset. This allows the algorithm to adapt online to realized temporal changes without the need for exact prior knowledge. The event trigger is based on probabilistic uniform error bounds used in Gaussian process regression. We derive regret bounds for adaptive resets without exact prior knowledge of the temporal changes and show in numerical experiments that ET-GP-UCB outperforms competing GP-UCB algorithms on both synthetic and real-world data. The results demonstrate that ET-GP-UCB is readily applicable without extensive hyperparameter tuning.

## 1 Introduction

Over the last two decades, Bayesian optimization (BO) has emerged as a powerful method for sequential decision-making and design of experiments under uncertainty (Garnett et al., 2010; Snoek et al., 2012; Calandra et al., 2016; Frazier & Wang, 2016; Chen et al., 2018; Neumann-Brosig et al., 2019; Griffiths & Hernández-Lobato, 2020; Colliandre & Muller, 2023). These problems are typically formalized as optimization problems of a *static* unknown objective function. At the core of BO is the exploration-exploitation trade-off, where the decision maker weights the potential benefit of a new unexplored query against the known reward of the currently assumed optimum. The performance of BO algorithms is typically measured in terms of regret, i.e., the difference between the actual decision taken and the (unknown) optimal one. If one assumes a static objective function, there exist algorithms that achieve a desirable sub-linear regret (see Garnett (2023) for an overview), meaning they will efficiently converge to the globally optimal solution.

In the real world, however, objective functions are often *time-varying*. Time-varying objective functions are widespread across various domains, including economics due to fluctuations in price or supply and demand (Primiceri, 2005; Dangl & Halling, 2012), environmental science (Marchant & Ramos, 2012; Marchant et al., 2014), and control systems due to changing system dynamics (Åström & Wittenmark, 1995; Ellis & Christofides, 2014; Colombino et al., 2019; Simonetto et al., 2020; König et al., 2021; Brunzema et al., 2022).

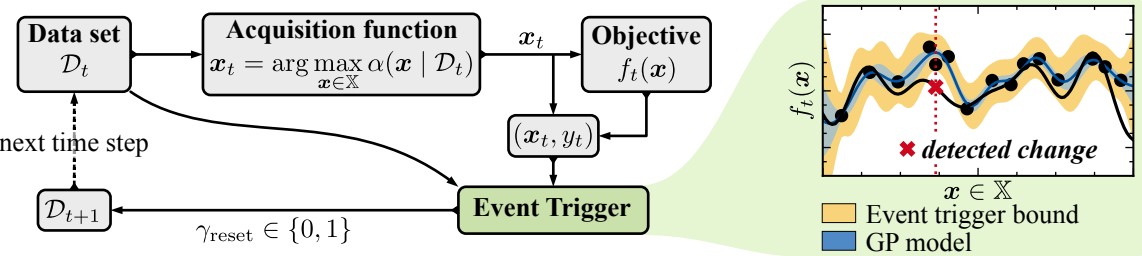

Figure 1: *Illustration of our proposed concept.* After the standard BO steps of optimizing the acquisition function and obtaining an observation, an *event trigger* decides whether to reset the data set ($\gamma_{\text{reset}} = 1$) if a change is detected (see red cross), or augment it with the observed data point in the usual way ($\gamma_{\text{reset}} = 0$). This event trigger allows our algorithm ET-GP-UCB to be adaptive to changes in the objective function without relying on exact prior knowledge of the rate of change.

A time-varying objective function alters the nature of the exploration-exploitation trade-off. Time affects the optimal decision and the information a decision maker gains from each query, and any collected data set becomes stale. Generally, in a setting of ongoing changes and without further assumptions, we cannot expect to find the optimal query, and conversely, no algorithm can achieve sub-linear regret (Besbes et al., 2015, Proposition 1). Instead, our goal is to design algorithms with an explicit relation between regret and the rate of change $\varepsilon$, which quantifies how fast the objective function varies. In their seminal work, Bogunovic et al. (2016) proposed two algorithms for the time-varying Bayesian optimization (TVBO) problem; however, the proposed algorithms and their subsequent variants rely on prior knowledge of the rate of change $\varepsilon$ as a hyperparameter. In typical BO applications, where the objective is unknown, knowledge of its rate of change $\varepsilon$ is hard to obtain. Moreover, in these algorithms, $\varepsilon$ is specified at the start and assumed to remain constant over time, which is rarely the case in practice.

To address these limitations, we propose the novel concept of event-triggered TVBO (see Figure 1). We model the objective function as time-invariant until an event trigger detects a significant deviation of the observed data from the current model. The algorithm then reacts to this change by resetting the data set. This way, the time-varying nature of the problem is considered in the algorithm's exploration-exploitation trade-off *only when necessary*. The event trigger replaces previously required exact prior knowledge of a rate of change, ET-GP-UCB exhibits strong empirical performance for different time variations, including gradual, sudden, and no change, and desirable theoretical properties under common regularity assumptions and upper and lower bounds on the rate of change. In summary, our contributions are as follows:

*(i)* A conceptually new event-triggered BO algorithm for TVBO;
*(ii)* Bayesian regret bounds for event-triggered BO for an unknown but bounded rate of change;
*(iii)* Empirical evaluations of ET-GP-UCB on synthetic data and real-world benchmarks.

## 2 Problem Setting

We aim to sequentially optimize an unknown time-varying objective function $f_t \colon \mathbb{X} \to \mathbb{R}$ on a compact set $\mathbb{X} \subset \mathbb{R}^d$ over the discrete time steps $t \in \mathbb{I}_{1:T} \coloneqq \{1, \dots, T\}$ up to the time horizon $T$ as

$$\boldsymbol{x}_t^* = \arg\max_{\boldsymbol{x} \in \mathbb{X}} f_t(\boldsymbol{x}). \tag{1}$$

Here and in the following, subscript $t$ denotes time dependency. At each time step, an algorithm chooses a query $\boldsymbol{x}_t \in \mathbb{X}$ and obtains noisy observations following the standard assumption in BO.

**Assumption 1** *Observations $y_t = f_t(\boldsymbol{x}_t) + w_t$ are perturbed by independent and identically distributed (i.i.d.) Gaussian noise with noise variance $\sigma_{\text{n}}^2$ as $w_t \sim \mathcal{N}(0, \sigma_{\text{n}}^2)$.*

Without assumptions on the temporal change, the problem is intractable. We impose the Markov chain model proposed by Bogunovic et al. (2016) and used in various prior applications (see Sec. 3).

**Assumption 2 (Bogunovic et al. (2016))** *Let $\mathcal{GP}(\mu, k(\boldsymbol{x}, \boldsymbol{x}'))$ be a Gaussian process (GP) with mean function $\mu: \mathbb{X} \to \mathbb{R}$ and kernel $k: \mathbb{X} \times \mathbb{X} \to \mathbb{R}$. Given i.i.d. samples $g_1, g_2, \dots$ with $g_i \sim \mathcal{GP}(0, k(\boldsymbol{x}, \boldsymbol{x}'))$, where $k(\boldsymbol{x}, \boldsymbol{x}')$ is a stationary kernel with $k(\boldsymbol{x}, \boldsymbol{x}') \leq 1$ for all $\boldsymbol{x}, \boldsymbol{x}' \in \mathbb{R}$, and a rate of change $\varepsilon \in [0, 1]$, the time-varying objective function $f_t(\boldsymbol{x})$ follows the Markov chain:*

$$f_t(\boldsymbol{x}) = \begin{cases} g_t(\boldsymbol{x}), & t = 1 \\ \sqrt{1 - \varepsilon} f_{t-1}(\boldsymbol{x}) + \sqrt{\varepsilon} g_t(\boldsymbol{x}), & t \geq 2. \end{cases} \tag{2}$$

**Remark 1** *Due to Assumption 2, any choice of $\varepsilon \in [0, 1]$ yields that $f_t \sim \mathcal{GP}(0, k)$ for all $t \in \mathbb{I}_{1:T}$.*

We further use standard smoothness assumptions on the kernel choice introduced in Srinivas et al. (2010) as well as Bogunovic et al. (2016).

**Assumption 3** *The kernel $k(\boldsymbol{x}, \boldsymbol{x}')$ is such that, given $f \sim \mathcal{GP}(0, k(\boldsymbol{x}, \boldsymbol{x}'))$, $L, L_f \geq 1$, and $a_0, b_0, a_1, b_1 \geq 0$, the following holds:*

$$\mathbb{P}\left\{ \sup_{\boldsymbol{x} \in \mathbb{X}} |f(\boldsymbol{x})| > L_f \right\} \leq a_0 e^{-(L_f/b_0)^2} \quad and \quad \mathbb{P}\left\{ \sup_{\boldsymbol{x} \in \mathbb{X}} \left| \frac{\partial f(\boldsymbol{x})}{\partial x_j} \right| > L \right\} \leq a_1 e^{-(L/b_1)^2}, j = 1, ..., d. \tag{3}$$

This assumption limits the choice of kernel to those that are at least four times differentiable (Ghosal & Roy, 2006, Theorem 5), such as the popular squared exponential (SE) kernel.

The performance of an algorithm optimizing (1) is measured in terms of the cumulative regret $R_T$.

**Definition 1** *Let $\boldsymbol{x}_t^*$ be the maximizer of $f_t(\boldsymbol{x})$, and $\boldsymbol{x}_t \in \mathbb{X}$ be the query at time step $t$. Then, the instantaneous regret at $t$ is $r_t = f_t(\boldsymbol{x}_t^*) - f_t(\boldsymbol{x}_t)$, and the regret after $T$ time steps is $R_T = \sum_{t=1}^{T} r_t$.*

Sub-linear regret is defined as $\lim_{T \to \infty} R_T/T = 0$. In the time-invariant setting ($\varepsilon = 0$), algorithms can achieve such sub-linear regret, e.g., GP-UCB (Srinivas et al., 2010, Theorem 2) with regret of order $\tilde{\mathcal{O}}(\sqrt{T})$[1]. For a time-varying objective function following Assumption 2, Bogunovic et al. (2016, Theorem 4.1) shows that the lower bound on the regret of an algorithm is linear as $\Omega(T\varepsilon)$.

**Problem Statement** We consider the time-varying optimization problem (1) under Assumptions 1–3. To develop a practical algorithm—and in contrast to the state-of-the-art in TVBO—we assume the exact rate of change $\varepsilon$ to be *unknown*. According to prior work (Bogunovic et al., 2016), it is reasonable to demand regret bounds that are no more than linear in $T$ and explicitly dependent on the true $\varepsilon$.

## 3 Related Work and Background

We build on prior work in BO and TVBO. This section gives an overview of these two fields.

**Bayesian Optimization** Our event trigger design and algorithm extends prior work by Srinivas et al. (2010) on the BO algorithm GP-UCB and transfers some of their concepts to the time-varying setting. Srinivas et al. (2010) proved sub-linear regret bounds when using an upper confidence bound (UCB) acquisition function of the form $\mu_{\mathcal{D}_t}(\boldsymbol{x}) + \sqrt{\beta_t} \sigma_{\mathcal{D}_t}(\boldsymbol{x})$, with standard posterior updates of the GP model, given a dataset $\mathcal{D}_t \coloneqq \{(\boldsymbol{x}_i, y_i)\}_{i=1}^{t-1}$, as

$$\underbrace{\mu_{\mathcal{D}_t}(\boldsymbol{x}) = \boldsymbol{k}_t(\boldsymbol{x})^T \left( \mathbf{K}_t + \sigma_n^2 \mathbf{I} \right)^{-1} \boldsymbol{y}_t}_{\text{time-invariant posterior mean}} \text{ and } \underbrace{\sigma_{\mathcal{D}_t}^2(\boldsymbol{x}) = k(\boldsymbol{x}, \boldsymbol{x}) - \boldsymbol{k}_t(\boldsymbol{x})^T \left( \mathbf{K}_t + \sigma_n^2 \mathbf{I} \right)^{-1} \boldsymbol{k}_t(\boldsymbol{x})}_{\text{time-invariant posterior variance}} \tag{4}$$

where $\mathbf{K}_t = [k(\boldsymbol{x}_i, \boldsymbol{x}_j)]_{i,j=1}^{t-1}$ is the Gram matrix, $\boldsymbol{k}_t(\boldsymbol{x}) = [k(\boldsymbol{x}_i, \boldsymbol{x})]_{i=1}^{t-1}$, and noisy measurements $\boldsymbol{y}_t = [y_1, \dots, y_{t-1}]^T$ follow Assumption 1. Our algorithm uses these standard posterior updates only as long as all the measurements comply with the current posterior model; it resets the dataset if this is no longer the case. Consequently, in contrast to GP-UCB, our algorithm can adapt to changes in the objective function by discarding stale data. For a detailed overview of other standard BO algorithms as well as the standard BO framework, we refer to Garnett (2023).

---

[1]As in Bogunovic et al. (2016) and Srinivas et al. (2010), we denote asymptotics up to log factors as $\tilde{\mathcal{O}}(\cdot)$.

**Time-Varying Bayesian Optimization** TVBO can be considered a special case of contextual BO (Krause & Ong, 2011), with the key difference that time is treated as a strictly increasing variable. The inclusion of time as an input for spatial-temporal monitoring within the BO framework was explored by Marchant & Ramos (2012) and Marchant et al. (2014). Building on spatio-temporal GPs, Bogunovic et al. (2016) introduced two time-varying UCB algorithms tailored for TVBO, where they suggest that changes in the objective function can be effectively captured using the model outlined in Assumption 2. Embedding this assumption into the GP surrogate model yields the following posterior updates

$$\underbrace{\tilde{\mu}_{\mathcal{D}_t}(\boldsymbol{x}) = \tilde{\boldsymbol{k}}_t(\boldsymbol{x})^T \left( \tilde{\mathbf{K}}_t + \sigma_{\mathrm{n}}^2 \mathbf{I} \right)^{-1} \boldsymbol{y}_t}_{\text{time-varying posterior mean}} \text{ and } \underbrace{\tilde{\sigma}_{\mathcal{D}_t}^2(\boldsymbol{x}) = k(\boldsymbol{x}, \boldsymbol{x}) - \tilde{\boldsymbol{k}}_t(\boldsymbol{x})^T \left( \tilde{\mathbf{K}}_t + \sigma_{\mathrm{n}}^2 \mathbf{I} \right)^{-1} \tilde{\boldsymbol{k}}_t(\boldsymbol{x})}_{\text{time-varying posterior variance}} \quad (5)$$

where $\tilde{\mathbf{K}}_t = \mathbf{K}_t \circ \mathbf{K}_t^{\text{time}}$ with $\mathbf{K}_t^{\text{time}} = [(1 - \varepsilon)^{|i-j|/2}]_{i,j=1}^{t-1}$, and $\tilde{\boldsymbol{k}}_t(\boldsymbol{x}) = \boldsymbol{k}_t(\boldsymbol{x}) \circ \boldsymbol{k}_t^{\text{time}}$ with $\boldsymbol{k}_t^{\text{time}} = [(1 - \varepsilon)^{|t+1-i|/2}]_{i=1}^{t-1}$, and $\circ$ is the Hadamard product. Using updates following (5) ensures that the posterior distribution follows the same dynamics as $f_t$. We will utilize this in the proof of regret bounds for algorithms without exact knowledge of $\varepsilon$ under the mentioned regularity assumptions.

Since the introduction of this setting by Bogunovic et al. (2016), their algorithm TV-GP-UCB with posterior updates as in (5) has been used in various applications, such as controller learning (Su et al., 2018), safe adaptive control (König et al., 2021), and hyperparameter optimization for reinforcement learning (Parker-Holder et al., 2020; 2021). Similarly, Brunzema et al. (2022) use time-varying posterior updates in their algorithm UI-TVBO but with a different temporal kernel. Both approaches rely on a priori estimation of the rate of change and assume that it stays constant over time. In contrast, the event trigger in ET-GP-UCB does not need an estimate of the rate of change and enables our algorithm to adapt to changes in the objective. We note that if the rate of change is known beforehand and the structural assumptions are satisfied, the TV-GP-UCB is an efficient choice since these properties are explicitly embedded in the algorithm.

Bogunovic et al. (2016) further introduced R-GP-UCB, which resets the data set periodically to account for data becoming uninformative over time. In essence, ET-GP-UCB is an adaptive version of R-GP-UCB and resets only if an obtained measurement is no longer consistent with the current dataset (cf. Sec. 5). As in R-GP-UCB, our algorithm uses time-invariant posterior updates within time intervals, but our time intervals do not have to be specified a priori based on $\varepsilon$; rather, they are determined online by the event trigger.

Under frequentist regularity assumption, Zhou & Shroff (2021) also propose an algorithm with a constant reset time, BR-GP-UCB[2], as well as a sliding window algorithm. The crucial difference to our problem setting is their assumption of a *fixed variational budget*, i.e., $\sum_{t=1}^{T} ||f_{t+1} - f_t||_{\mathcal{H}} < B_T$, where $\mathcal{H}$ is a Reproducing Kernel Hilbert Space (RKHS) and $B_T$ a scalar. Leveraging these same assumptions, Deng et al. (2022) propose an algorithm, W-GP-UCB, which utilizes a similar weighted GP model to TV-GP-UCB. In this setting of a fixed variational budget, sub-linear regret is possible (Besbes et al., 2015). In contrast, we are interested in the setting given by Assumption 2, which corresponds to a variational budget that increases linearly in time; hence, the best possible regret is also linear (Besbes et al., 2015, Proposition 1). Nevertheless, in Appendix A.5, we empirically compare ET-GP-UCB against the algorithms of Zhou & Shroff (2021) and Deng et al. (2022) and display improved performance on their benchmarks with less prior information.

**Event-Triggered Learning** The idea of resetting a dataset only if an important change occurs as detected by a trigger event is similar to the concepts of event-triggered learning (Solowjow & Trimpe, 2020) and event-triggered online learning (Umlauft & Hirche, 2019). In the multi-armed bandit setting, Wei & Luo (2021) discuss the idea of including tests into optimization algorithms to detect and adapt to non-stationarity. Our algorithm ET-GP-UCB is the first to introduce event triggers to BO and specifically TVBO. Since then, this idea has been taken up for a different BO problem, namely safe BO, in recent work (Holzapfel et al., 2024). The event-triggered concepts aim to be efficient with the available data, only updating a model when necessary or only adding new data that is relevant. ET-GP-UCB also aims for data efficiency; as long as we have a good model of our objective function, we want to use already available data to minimize regret. If the event trigger detects a significant model mismatch, we reset the dataset and thereby delete stale data.

---

[2]We label it BR-GP-UCB to avoid confusion with R-GP-UCB of Bogunovic et al. (2016).

**Dynamic and Non-Stationary Multi-Armed Bandits**  Related problems to (1) have also been considered in the non-stationary multi-armed bandit (MAB) literature. The key distinction to our problem setting is that for MABs the number of arms is finite and they are not correlated. Similar to our setting with ongoing changes, Whittle (1988) introduce restless MABs; here, the state of an arm can change in each step according to a known (yet arbitrary) stochastic transition function (Slivkins & Upfal, 2008). Such restless MABs are known to be intractable (Papadimitriou & Tsitsiklis, 1994; Slivkins & Upfal, 2008). For restless bandits with a "time-variation parameter" greater zero linear regret as $\Omega(T)$ is inevitable (Slivkins & Upfal, 2008, Theorem 1.3). The setting in Slivkins & Upfal (2008) is, in its effects, similar to the regularity assumption on the temporal changes in Assumption 2 with a corresponding lower bound on the regret of $\Omega(T\varepsilon)$ (Bogunovic et al., 2016, Theorem 4.1). Also related are MABs under a fixed number of abrupt distribution shifts (Auer et al., 2019; Abbasi-Yadkori et al., 2023). For these problems, adaptive resetting algorithms have been proposed (Chen et al., 2019; Wei & Luo, 2021; Suk & Kpotufe, 2022). In this more structured problem setting, similar to the frequentist regularity assumptions discussed above, sub-linear regret is possible.

## 4 Regret Bounds for TVBO with Adaptive Resets

The core goal of this work is to develop an algorithm for an unknown rate of change. To achieve this goal, we first generalize the results by Bogunovic et al. (2016) and derive regret bounds for a BO algorithm with variable resets. We analyze the general problem for all strategies that reset anytime between a lower ($\underline{N}$) and upper ($\bar{N}$) reset threshold. Our analysis reveals that for any resetting strategy, given reasonable $\underline{N}$ and $\bar{N}$, linear regret with explicit dependency on the true $\varepsilon$ is guaranteed, even if the exact $\varepsilon$ is unknown to the resetting algorithm. This is in contrast to previous work on TVBO, where the true $\varepsilon$ does not only appear in the regret bounds but needs to be known to the algorithm or GP model. We first introduce a theorem applicable to all algorithms with such a general resetting strategy, and then apply it to specific kernels to derive scaling laws. In Sec. 5, we will propose a specific event trigger as a resetting strategy.

**Theorem 1** *Let the domain $\mathbb{X} \subset [0, r]^d \subset \mathbb{R}^d$ be convex and compact with $d \in \mathbb{N}_+$ and let $f_t$ follow Assumption 2 with a kernel $k(\boldsymbol{x}, \boldsymbol{x}')$ such that Assumption 3 is satisfied. Pick $\delta \in (0, 1)$ and set $\beta_t = 2 \ln \left(2\pi^2 t^2/(3\delta)\right) + 2d \ln \left(t^2 db_1 r \sqrt{\ln(2da_1\pi^2 t^2/(3\delta))}\right)$. Pick a lower block length $\underline{N} \in \mathbb{N}_+$ and a upper block length $\bar{N} \in \mathbb{N}_+$ with $\underline{N} \leq \bar{N} \leq T$. Let $\gamma_{\bar{N}}$ be the maximum information gain over $\bar{N}$ points. Then, running any algorithm with block sizes between $\underline{N}$ and $\bar{N}$ satisfies*

$$R_T \leq \sqrt{C_1 T \beta_T \left(\frac{T}{\underline{N}} + 1\right) \gamma_{\bar{N}}} + 2 + T\phi_T(\varepsilon, \bar{N}) \tag{6}$$

*with probability at least $1 - \delta$, where $C_1 = 8/\ln(1 + \sigma_{\mathrm{n}}^{-2})$ and we defined*

$$\phi_T(\varepsilon, \bar{N}) := 2\sqrt{\beta_T(3\sigma_{\mathrm{n}}^{-2} + \sigma_{\mathrm{n}}^{-4})\bar{N}^3\varepsilon} + 2(\sigma_{\mathrm{n}}^{-2} + \sigma_{\mathrm{n}}^{-4})\bar{N}^3\varepsilon(b_0 + \sqrt{2}\sigma_{\mathrm{n}})\sqrt{\ln \frac{4(a_0+1)\pi^2 T^2}{3\delta}}. \tag{7}$$

**Sketch of proof:**  We extend the analysis of R-GP-UCB to a setting in which resets are not enforced periodically but can occur at any time step between $\underline{N}$ and $\bar{N}$. Building on Assumptions 1 to 2, we use the upper threshold $\underline{N}$ to bound the difference between the model used in the algorithm (time-invariant model, see (4)) and the time-varying model in (5). We use the lower threshold $\bar{N}$ to bound the maximum number of resets over the time horizon. The full proof is in Appendix F. □

We next apply this theorem to specific kernels. We specify how $\underline{N}$ and $\bar{N}$ scale with upper and lower bounds on the rate of change to obtain the desirable scaling of the regret.

**Corollary 1** *Let the assumptions in Theorem 1 hold. Given a lower and upper bound on the true $\varepsilon$ such that $\varepsilon \in [\underline{\varepsilon}, \bar{\varepsilon}] \subset (0, 1)$ there exist $\lambda_1 \in (0, 1]$ and $\lambda_2 \in [1, \frac{1}{\varepsilon})$ such that $\underline{\varepsilon} = \lambda_1 \cdot \varepsilon$ and $\bar{\varepsilon} = \lambda_2 \cdot \varepsilon$. For the $\underline{SE\ kernel}$, set $\underline{N} = \Theta(\min\{T, \bar{\varepsilon}^{-1/4}\})$ and $\bar{N} = \Theta(\min\{T, \underline{\varepsilon}^{-1/4}\})$. Then, with probability $1 - \delta$, we have*

$$R_T = \tilde{\mathcal{O}}\left(\max\left\{\sqrt{T},\ \lambda_2^{1/8}\varepsilon^{1/8}T,\ \lambda_1^{-3/8}\varepsilon^{1/8}T,\ \lambda_1^{-3/4}\varepsilon^{1/4}T\right\}\right). \tag{8}$$

*For the Matèrn kernel ($\nu > 2$), define $\xi = \frac{d(d+1)}{2\nu + d(d+1)}$ and set $\underline{N} = \Theta(\min\{T, \bar{\varepsilon}^{-1/(4-\xi)}\})$ and $\bar{N} = \Theta(\min\{T, \underline{\varepsilon}^{-1/(4-\xi)}\})$. Then, with probability $1 - \delta$, we have*

$$R_T = \tilde{\mathcal{O}}\left(\max\left\{\sqrt{T^{1-\xi}}, \lambda_1^{-\frac{\xi}{2(4-\xi)}}\lambda_2^{\frac{1}{2(4-\xi)}}\varepsilon^{\frac{1-\xi}{2(4-\xi)}}T, \lambda_1^{-\frac{3}{2(4-\xi)}}\varepsilon^{\frac{1-\xi}{2(4-\xi)}}T, \lambda_1^{-\frac{3}{4-\xi}}\varepsilon^{\frac{1-\xi}{4-\xi}}T\right\}\right). \tag{9}$$

From Corollary 1, we can derive the following insights: *(i)* It is possible to design algorithms with regret explicitly depending on $\varepsilon$, without knowing its exact value. *(ii)* A larger interval for $\varepsilon$, i.e., decreasing $\lambda_1$ or increasing $\lambda_2$, results in a worse scaling of the regret bound. It essentially is the price one has to pay for not knowing the exact value of $\varepsilon$. *(iii)* For $\varepsilon = \underline{\varepsilon} = \bar{\varepsilon}$, hence $\lambda_1 = \lambda_2 = 1$, the scaling of R-GP-UCB is recovered. From Theorem 1, we can further observe that for $\varepsilon = 0$ the bounds recover the sub-linear scaling of GP-UCB iff an appropriate lower bound on $\varepsilon$ is specified such that $\underline{N} = T$ which follows intuition as no resets should should take place.

The above results demonstrate that it is possible to design algorithms that do not require the exact rate of change; as long as lower and upper bounds on $\varepsilon$ are known and resets occur within $[\bar{N}, \underline{N}]$ the algorithm has bounded regret. Crucially, we did not actually specify *when* to reset, thus our analysis includes random and "worst case" resets. Next, we utilize the flexibility in choosing the reset timing within $[\bar{N}, \underline{N}]$ to achieve better empirical regret, e.g., as compared to periodic resets.

## 5 An Event Trigger for Adaptive Resets in TVBO

We will now introduce our algorithm ET-GP-UCB. It will decide *when* to reset through its event trigger (see Figure 1). We build on established uniform error bounds for GP regression to decide if new data is consistent with the posterior. When new data deviates significantly, we reset the dataset. This idea constitutes a new approach compared to previous work in TVBO, as the resulting algorithm does not rely on the true rate of change $\varepsilon$ as a hyperparameter. The core component of our algorithm is the event trigger:

**Definition 2 (Event trigger in TVBO)** *Given a test function $\psi_t$ and a threshold function $\kappa_t$, both of which can depend on the current dataset $\mathcal{D}_t$ and the latest query location and measurement pair $(\boldsymbol{x}_t, y_t)$, the event trigger at time $t$ is*

$$\gamma_{\text{reset}} = 1 \iff \psi_t\left(\mathcal{D}_t, (\boldsymbol{x}_t, y_t)\right) > \kappa_t\left(\mathcal{D}_t, (\boldsymbol{x}_t, y_t)\right) \tag{10}$$

*where $\gamma_{\text{reset}}$ is the binary indicator for whether to reset the dataset ($\gamma_{\text{reset}} = 1$) or not ($\gamma_{\text{reset}} = 0$).*

We next design and develop $\psi_t$ and $\kappa_t$ for the specific event trigger of ET-GP-UCB. We base these functions on Bayesian uniform error bounds for GP regression (Srinivas et al., 2010). Other design options would also be possible, e.g., likelihood-based or performance-based event triggers. In the following, we denote the time step of a reset as $t_r^{(1)}, t_r^{(2)}, \ldots$ and define $t_r \in \left[t_r^{(i)}, t_r^{(i+1)}\right] \subset \mathbb{N}_+$ as the time in between resets. We have $t_r = t$ if no reset has yet occurred.

**Lemma 1 (Srinivas et al. (2010, Lemma 5.5))** *Pick $\delta \in (0, 1)$ and set $\rho_{t_r} = 2\ln\frac{\pi_{t_r}}{\delta}$, where $\sum_{t_r \geq 1}\pi_{t_r}^{-1} = 1$, $\pi_{t_r} > 0$. Let $f \sim \mathcal{GP}(0, k(\boldsymbol{x}, \boldsymbol{x}'))$. Then, $|f(\boldsymbol{x}_t) - \mu_{\mathcal{D}_t}(\boldsymbol{x}_t)| \leq \sqrt{\rho_{t_r}}\sigma_{\mathcal{D}_t}(\boldsymbol{x}_t)$ holds for all $t_r \geq 1$ with probability at least $1 - \delta$.*

**Remark 2** *For $\sum_{t_r \geq 1}\pi_{t_r}^{-1} = 1$ to hold, one can choose $\pi_{t_r} = \pi^2 t_r^2/6$ as in Srinivas et al. (2010).*

The result gives a high probability bound for all time steps on the absolute deviation between the posterior mean and a time-invariant objective function at the query location $\boldsymbol{x}_t$ based on the posterior variance and parameter $\rho_{t_r}$. If a time-varying objective function satisfies this bound for all time steps, the time variations are negligible with high probability (i.e., $\varepsilon \approx 0$). On the other hand, if the bound in Lemma 1 is violated at a chosen query location $\boldsymbol{x}_t$, the objective function has likely changed.

The algorithm needs to evaluate the trigger event at every time step. While it cannot directly evaluate $|f_t(\boldsymbol{x}_t) - \mu_{\mathcal{D}_t}(\boldsymbol{x}_t)|$ in Lemma 1 (cf. Assumption 1), we can evaluate $|y_t - \mu_{\mathcal{D}_t}(\boldsymbol{x}_t)|$. Thus, we adapt Lemma 1 to design a probabilistic bound that includes only $y_t$ instead of $f_t(\boldsymbol{x}_t)$. For this, we first bound the noise in probability. All proofs of the following Lemmas are in Appendix G.

---

**Algorithm 1** Event-triggered GP-UCB (ET-GP-UCB).

---

1: **Define:** $\mathcal{GP}(0, k)$, $\mathbb{X} \subset \mathbb{R}^d$, $\delta_{\mathrm{B}} \in (0, 1)$, $\mathcal{D}_1 = \emptyset$, lower reset bound $\underline{N}$, upper reset bound $\bar{N}$, $t_r = 1$
2: **for** $t = 1, 2, \ldots, T$ **do**
3:      Train GP model with the current data set $\mathcal{D}_t$
4:      Choose $\beta_t$ (e.g., according to Theorem 1)
5:      Select $\boldsymbol{x}_t = \arg \max_{\boldsymbol{x} \in \mathbb{X}} \mu_{\mathcal{D}_t}(\boldsymbol{x}) + \sqrt{\beta_t} \sigma_{\mathcal{D}_t}(\boldsymbol{x})$          $\triangleright$ time-invariant posterior using (4)
6:      Sample next observation $y_t = f_t(\boldsymbol{x}_t) + w_t$
7:      $\gamma_{\mathrm{reset}} \leftarrow$ EVENT TRIGGER$(\mathcal{D}_t, (y_t, \boldsymbol{x}_t), \delta_{\mathrm{B}})$          $\triangleright$ evaluate (10) with (11)
8:      **if** ($\gamma_{\mathrm{reset}}$ **and** $t_r \in [\underline{N}, \bar{N}]$) **or** $t_r = \bar{N}$ **then**      $\triangleright$ reset only if within reset window
9:          Reset dataset $\mathcal{D}_{t+1} = \{(y_t, \boldsymbol{x}_t)\}$ and set $t_r = 1$
10:      **else**
11:          Update dataset $\mathcal{D}_{t+1} = \mathcal{D}_t \cup \{(y_t, \boldsymbol{x}_t)\}$ and set $t_r = t_r + 1$

---

**Lemma 2** *Pick $\delta \in (0, 1)$ and set $\bar{w}_{t_r}^2 = 2\sigma_{\mathrm{n}}^2 \ln \frac{\pi_{t_r}}{\delta}$, where $\sum_{t_r \geq 1} \pi_{t_r}^{-1} = 1$, $\pi_{t_r} > 0$. Then, the noise sequence in Assumption 1 obtained since the last reset satisfies $|w_{t_r}| \leq \bar{w}_{t_r}$ for all $t_r \geq 1$ with probability at least $1 - \delta$.*

With Lemma 2, we can extend Lemma 1 to build a probabilistic error bound that can act as the test and corresponding threshold functions of our event trigger.

**Lemma 3** *Let $f_t(\boldsymbol{x})$ follow Assumption 2 with $\varepsilon = 0$. Pick $\delta_{\mathrm{B}} \in (0, 1)$ and set $\rho_{t_r} = 2 \ln \frac{2\pi_{t_r}}{\delta_{\mathrm{B}}}$, where $\sum_{t_r \geq 1} \pi_{t_r}^{-1} = 1$, $\pi_{t_r} > 0$. Also set $\bar{w}_{t_r}^2 = 2\sigma_{\mathrm{n}}^2 \ln \frac{2\pi_{t_r}}{\delta_{\mathrm{B}}}$. Then, observations $y_t$ under Assumption 1 satisfy $|y_t - \mu_{\mathcal{D}_t}(\boldsymbol{x}_t)| \leq \sqrt{\rho_{t_r}} \sigma_{\mathcal{D}_t}(\boldsymbol{x}_t) + \bar{w}_{t_r}$ for all $t_r \geq 1$ with probability at least $1 - \delta_{\mathrm{B}}$.*

We can now formally define the event trigger for ET-GP-UCB following the framework outlined in Definition 2. Leveraging the result from Lemma 3, we choose as the test and threshold function

$$\underbrace{\psi_t\left(\mathcal{D}_t, (\boldsymbol{x}_t, y_t)\right) = |y_t - \mu_{\mathcal{D}_t}(\boldsymbol{x}_t)|}_{:= \text{ test function } \psi_t \text{ in Def. 2}} \text{ and } \underbrace{\kappa_t\left(\mathcal{D}_t, (\boldsymbol{x}_t, y_t)\right) = \sqrt{\rho_{t_r}} \sigma_{\mathcal{D}_t}(\boldsymbol{x}_t) + \bar{w}_{t_r}}_{:= \text{ threshold function } \kappa_t \text{ in Def. 2}}, \tag{11}$$

respectively. With this, we have designed a principled event trigger to detect changes in the objective function. Note that our event trigger monitors for changes in the objective function only at the current query location and cannot detect changes anywhere else. Our problem setting restricts an algorithm to querying the objective only once per time step. The resulting algorithm ET-GP-UCB is then as follows: We perform standard GP-UCB as long as the event trigger evaluates to $\gamma_{\mathrm{reset}} = 0$. When the event trigger is activated the objective function $f_t$ has likely changed, and the data has become stale. If $t_r \in [\underline{N}, \bar{N}]$, ET-GP-UCB resets the dataset and restarts the optimization. This is summarized in Algorithm 1. The single design parameter of our event trigger is $\delta_{\mathrm{B}}$ in Lemma 3, which has a well-understood meaning as defining the tightness of the probabilistic uniform error bound. Since resets are confined to the range between the lower and upper reset thresholds, ET-GP-UCB guarantees a regret no worse than that specified in Corollary 1.

## 6 Empirical Evaluations

In the following, we will first investigate the empirical impact of $\underline{N}$ and $\bar{N}$ on the reset times of the algorithm as well as on the regret. In Section 6.2, we will benchmark our algorithm against other baselines on various synthetic and real-world problems and end with practical extensions for our proposed algorithm.

### 6.1 Empirical Impact of $\underline{N}$ and $\bar{N}$ on Reset Times and Regret

**Impact on the Reset Times**    To analyze the impact of $\underline{N}$ and $\bar{N}$ on the behavior and performance of ET-GP-UCB, we first define the window size in which the event trigger can reset the data set as $W := \bar{N} - \underline{N}$. Further, note that the time of reset using (11) in the event trigger framework is a random variable as it depends on the realization of $f_t$ as well as the observation model. The ridgeline plot in Figure 2 (left) shows the empirical distribution of this reset time as (scaled) histograms; starting from the reset time of R-GP-UCB, the window size increases. For this, we ran ET-GP-UCB on 30 000 different objective functions following Assumption 2, so-called within-model comparisons (cf. Section 6.2.1), for each window size. Small window sizes cut off the probability mass in the tails of the histograms, while for larger window sizes, the histogram resembles an inverse gamma distribution.

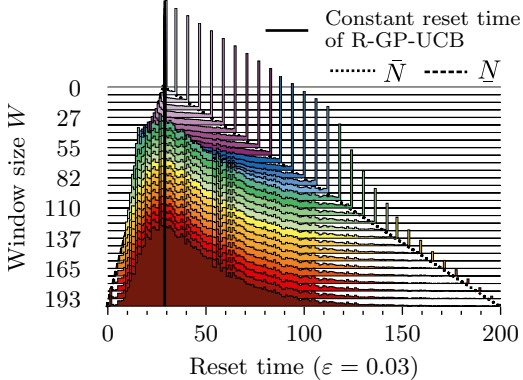 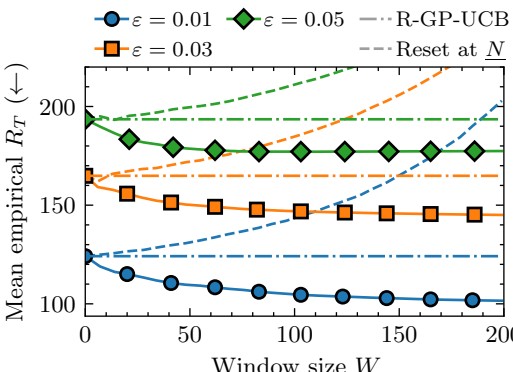

Figure 2: *Left:* Influence of the window size on the empirical PDF of reset times. For small window sizes, the probability mass in the tails is cut off through $\underline{N}$ and $\bar{N}$. As the window size $W$ increases, hence the flexibility of our event trigger increases, the histogram resembles an inverse-gamma-like distribution. *Right:* Influence of the window size on the empirical regret for different $\varepsilon$. As the window size increases, the regret of ET-GP-UCB (markers) decreases always staying below R-GP-UCB (dash-dotted lines). Increasing the flexibility of our trigger empirically reduces regret. Always resetting at $\underline{N}$ (dashed line) shows increased empirical regret. This indicates that Corollary 1 is a relatively loose bound for our trigger.

**Impact on the Regret** Figure 2 (right) demonstrates how increasing the window size affects regret for different $\varepsilon$. As the window size grows, ET-GP-UCB's performance (lines with markers) consistently remains below the regret of R-GP-UCB with known $\varepsilon$ (dash-dotted lines). This suggests that even coarse bounds lead to robust performance, which is desirable from a practical viewpoint. The improved performance is contrary to the regret bounds in Corollary 1, which gets larger as the window size grows. However, Corollary 1 gives regret bounds for *all* triggers, including the worst-case. Empirically, the proposed event trigger performs much better than the worst-case and even improves on R-GP-UCB despite less prior knowledge of the rate of chance. ET-GP-UCB performs best for larger window sizes. This suggests that prior knowledge of the rate of change is less important than the regret bounds in Corollary 1 imply. The proposed event trigger uses the gained flexibility from the theoretical investigation to perform resets *when necessary* within the reset window. Other event triggers may not show the same behavior: resetting, e.g., always at the lower bound $\underline{N}$ (dashed lines in Figure 2) exhibits increased regret with growing window sizes highlighting the importance of choosing a suitable resetting strategy for good empirical performance.

## 6.2 Empirical Evaluation on Synthetic and Real-World Data

Using both synthetic data and standard BO benchmarks, we compare ET-GP-UCB to state-of-the-art GP-UCB baseline for TVBO. Specifically, we compare against R-GP-UCB, which resets periodically, TV-GP-UCB (Bogunovic et al., 2016), which gradually forgets past data points, and UI-TVBO (Brunzema et al., 2022), which continuously injects uncertainty at past data points. We further compare these time-varying algorithms to the time-invariant GP-UCB (Srinivas et al., 2010). In our experiments, we find that:

*(i)* The adaptive resetting of ET-GP-UCB always outperforms periodic resetting;
*(ii)* ET-GP-UCB outperforms all baselines on synthetic data if the rate of change is misspecified;
*(iii)* ET-GP-UCB outperforms all baselines on real-world data (where the rate of change is unknown).

We reiterate that ET-GP-UCB does not rely on knowledge about the true rate of change. To highlight the generalization capabilities of ET-GP-UCB to different scenarios, we use the same parameter for the event trigger ($\delta_{\mathrm{B}} = 0.1$ in Lemma 3) in all experiments. A sensitivity analysis of the hyperparameter $\delta_{\mathrm{B}}$ on the regret is in Appendix B and shows that ET-GP-UCB performs similarly for a wide range of $\delta_{\mathrm{B}}$.

As in (Bogunovic et al., 2016) and (Srinivas et al., 2010), we utilize a logarithmic scaling $\beta_t$ as $\beta_t = \mathcal{O}(d\ln(t))$, where $\beta_t = c_1 \ln(c_2 t)$. This approximates $\beta_t$ in Theorem 1 and allows for a direct comparison to Bogunovic et al. (2016), as they suggest to set $c_1 = 0.8$ and $c_2 = 4$ for a suitable exploration-exploitation trade-off. All experiments in this section are conducted using BoTorch (Balandat et al., 2020) and GPyTorch (Gardner

Table 1: Within-model comparisons. *Top:* Algorithms with time-invariant updates following (4). *Bottom:* Algorithms with more compute intensive time-varying updates similar to (5). The best median performance for each section is highlighted in **bold**, and in  green  for the best performance across top and bottom.

| Algorithm | $\varepsilon = 0.01$ $R_T/T$ ($\downarrow$) | $\varepsilon = 0.03$ $R_T/T$ ($\downarrow$) | $\varepsilon = 0.05$ $R_T/T$ ($\downarrow$) | Miss. $\varepsilon = 0.001^{\dagger}$ $R_T/T$ ($\downarrow$) | Miss. $\varepsilon = 0.2^{\dagger}$ $R_T/T$ ($\downarrow$) |
|---|---|---|---|---|---|
| GP-UCB | $0.748^{0.904}_{0.610}$ | $1.051^{1.243}_{0.924}$ | $1.276^{1.377}_{1.088}$ | $1.276^{1.377}_{1.088}$ | $1.276^{1.377}_{1.088}$ |
| R-GP-UCB | $0.622^{0.681}_{0.571}$ | $0.831^{0.894}_{0.770}$ | $0.985^{1.035}_{0.899}$ | $0.902^{0.985}_{0.829}$ | $1.054^{1.119}_{0.995}$ |
| ET-GP-UCB$^{0.05}_{0.01}$* | $0.604^{0.682}_{0.531}$ | $0.778^{0.841}_{0.720}$ | $0.879^{0.966}_{0.824}$ | $0.879^{0.966}_{0.824}$ | $0.879^{0.966}_{0.824}$ |
| ET-GP-UCB$^{0.1}_{0.001}$* | $0.507^{0.581}_{0.448}$ | $0.688^{0.782}_{0.654}$ | $0.866^{0.927}_{0.806}$ | $0.866^{0.927}_{0.806}$ | $0.866^{0.927}_{0.806}$ |
| ET-GP-UCB$^{1.0}_{0.0}$* | $\mathbf{0.483^{0.571}_{0.407}}$ | $\mathbf{0.686^{0.738}_{0.639}}$ | $\mathbf{0.849^{0.899}_{0.748}}$ | $\mathbf{0.849^{0.899}_{0.748}}$ | $\mathbf{0.849^{0.899}_{0.748}}$ |
| TV-GP-UCB | $\mathbf{0.299^{0.365}_{0.258}}$ | $\mathbf{0.500^{0.565}_{0.447}}$ | $\mathbf{0.622^{0.686}_{0.582}}$ | $0.960^{1.036}_{0.854}$ | $1.223^{1.289}_{1.126}$ |
| UI-TVBO | $0.351^{0.376}_{0.311}$ | $0.647^{0.678}_{0.600}$ | $0.868^{0.904}_{0.833}$ | $\mathbf{0.953^{1.072}_{0.839}}$ | $1.381^{1.417}_{1.340}$ |

\* Our algorithms with formatting ET-GP-UCB $^{\text{upper bound on }\varepsilon}_{\text{lower bound on }\varepsilon}$. $^{\dagger}$ True rate of change is $\varepsilon = 0.05$.

et al., 2018), and full details are in Appendix A. All results show the median and interquartile performance. In the following, we indicate the direction of better performance with upward ($\uparrow$) and downward ($\downarrow$) arrows.

### 6.2.1 Within-Model Comparison

First, we consider the same two-dimensional setting for synthetic data as in Bogunovic et al. (2016). We choose the compact set to be $\mathbb{X} = [0, 1]^2$ and generate 50 different objective functions according to Assumption 2 with a known $\varepsilon$, a SE kernel with fixed lengthscales $l = 0.2$, time horizon $T = 400$, and a noise variance of $\sigma_{\mathrm{n}}^2 = 0.02$. The setting in which all the hyperparameters are known to the algorithm is referred to as *within-model comparison* (Hennig & Schuler, 2012); such comparisons are used to evaluate an algorithm's performance given all assumptions are fulfilled (cf. Appendix A.1 for further details on the creation of the within-model objective functions). Here, TV-GP-UCB acts as the reference case with full information. It shows the best performance an UCB algorithm can obtain in this setting as it takes Assumption 2 explicitly into account by using the time-varying posterior updates in (5). R-GP-UCB is also parameterized using the true $\varepsilon$ for the periodic reset time as $N_{\mathrm{const}} = \lceil \min\{T, 12\varepsilon^{-1/4}\} \rceil$ (Bogunovic et al., 2016, Corollary 4.1). For ET-GP-UCB, we compare three different variants with different lower and upper bounds on $\varepsilon$ and set $\bar{N} = \lceil \min\{T, 12\underline{\varepsilon}^{-1/4}\} \rceil$ and $\underline{N} = \lceil \min\{T, 12\bar{\varepsilon}^{-1/4}\} \rceil$ according to Corollary 1. For UI-TVBO, we set $\hat{\sigma}_w^2 = \varepsilon$ (cf. Brunzema et al. (2022)). Table 1 shows the normalized regret (median$^{q75\%}_{q25\%}$) for different $\varepsilon$. We can observe that in all cases the ET-GP-UCB variant with the largest flexibility of the event trigger outperforms all baselines with a time-invariant GP model (top). As expected, TV-GP-UCB is the best-performing algorithm if the rate of change is correct. However, if the rate of change is misspecified (right columns), ET-GP-UCB also outperforms the baselines with a time-varying GP model as it does not rely on exact knowledge of $\varepsilon$ and can react to the *realized* changes of $f_t$.

### 6.2.2 Real-World Temperature Data

To benchmark the algorithms on real-world data, we use the temperature dataset collected from 46 sensors deployed at Intel Research Berkeley over eight days at 10 minute intervals.[3] This dataset was also used as a benchmark in previous work on GP-UCB and TVBO (Srinivas et al., 2010; Krause & Ong, 2011; Bogunovic et al., 2016). The objective is to activate the sensor with the highest temperature at each time step resulting in a spatio-temporal monitoring problem. In our TVBO setting, $f_t$ consists of all sensor readings at time step $t$, and an algorithm can query only a single sensor every 10 minutes. Hence, the regret following Definition 1 is the difference between the maximum and the recorded temperature by the activated sensor.

For each experiment, two test days are selected, while the preceding days serve as the training set. Using this training set, all data is normalized to have a mean of zero and a standard deviation of one. The empirical covariance matrix is computed based on the training set and employed as the kernel for all algorithms as in Bogunovic et al. (2016). The sensor data of the days 7 and 8 is shown in Figure 3 (a) and an overview of the remaining days is given in Appendix E. For TV-GP-UCB and UI-TVBO, we set $\varepsilon = \hat{\sigma}_w^2 = 0.03$ and the

---

[3]We thank the researchers at Intel for the publically available data set under: https://db.csail.mit.edu/labdata/labdata.html.

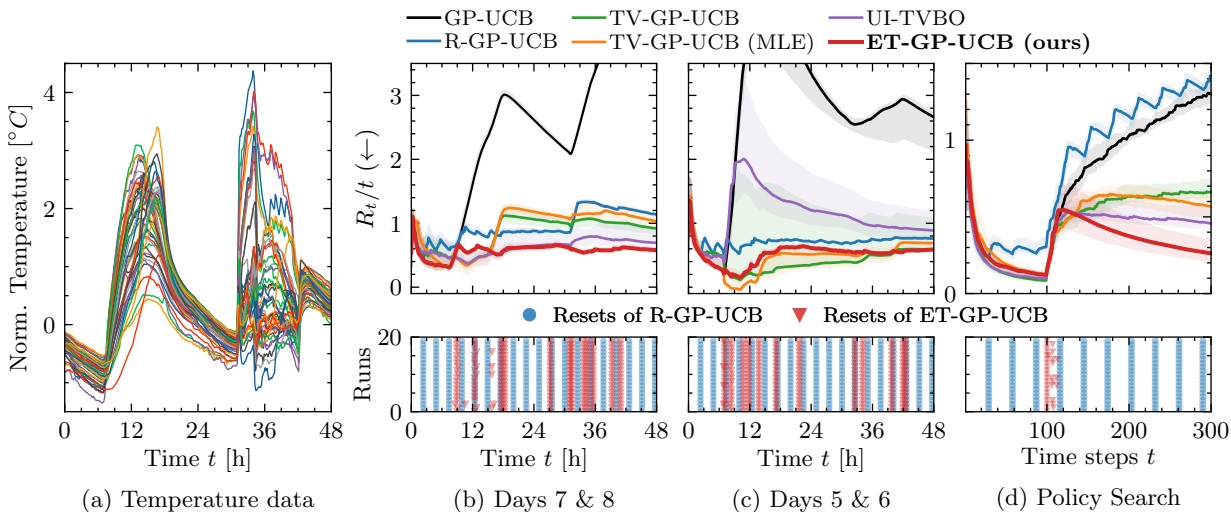

Figure 3: *Real-world examples.* Subfigure (a) shows the temperature data from the test days used in Bogunovic et al. (2016). In (b), the performance of the algorithms on these two days (top) and the resets of R-GP-UCB and ET-GP-UCB (bottom) are displayed. Subfigure (c) shows the performance on other test days. ET-GP-UCB displays superior performance improvement compared to all other algorithms. For these experiments on real-world data, we use the same parameter for the event trigger as in all previous experiments whereas TV-GP-UCB and R-GP-UCB relied on estimating $\varepsilon$. Lastly, in subfigure (d), ET-GP-UCB also outperforms all competing baselines on a policy search benchmark. For all methods, the median performance and the interquartile (25% and 75%) are displayed.

periodic reset time of R-GP-UCB to $N_{\text{const}} = 15$ according to Bogunovic et al. (2016). For this benchmark, we further compare against TV-GP-UCB with online maximum likelihood estimation (MLE) of $\varepsilon$ as one might choose to do in practice if the true rate of change is unknown. Note that this significantly increases the computational overhead. For ET-GP-UCB, we use the best-performing variant form Sec. 6.2.1 as well as the same parameter for the event trigger as in all the previous experiments.

Figure 3 (b) presents the normalized regret (top) for each algorithm over 20 runs and the resets of R-GP-UCB and ET-GP-UCB (bottom). In this real-world benchmark, ET-GP-UCB outperforms all the competing baselines. It highlights that using a constant rate of change in the GP model as in TV-GP-UCB can be problematic for real-world problems. ET-GP-UCB also shows the best performance on other test days as shown in Figure 3 (c). considering the larger interquartile of TV-GP-UCB. While ET-GP-UCB and TV-GP-UCB have similar median performance, ET-GP-UCB displayed a significantly smaller interquartile range, indicating more consistent and reliable results. From the bottom figure, we can observe the adaptive nature of ET-GP-UCB. Considering the timing of resets of ET-GP-UCB, it appears that especially when changes in the objective function are more distinct (e.g., changes between night and day), the proposed event trigger framework is empirically beneficial compared to using an estimated constant rate of change.

### 6.2.3 Policy Search in Time-Varying Environments

We also compare all algorithms on a four-dimensional policy search benchmark of a cart-pole system as proposed in (Brunzema et al., 2022). Given the system dynamics of the cart-pole system, it is possible to calculate the optimal policy and thus calculate the regret following Definition 1. To induce a change in the objective function, we increase the friction in the bearing at $t = 100$ by a factor of 2.5. We expect the event trigger of ET-GP-UCB to detect this change in the objective function and converge to the new optimum after resetting the dataset. In contrast to previous examples, we also perform online hyperparameter optimization of the spatial lengthscales using gamma priors to guide the search (all parameters are listed in Appendix H). For TV-GP-UCB, we choose $\varepsilon = 0.03$ and we pick the constant reset timing of R-GP-UCB according to Bogunovic et al. (2016, Corollary 4.1). For ET-GP-UCB, we again choose the same parameter for the event trigger as in all previous examples.

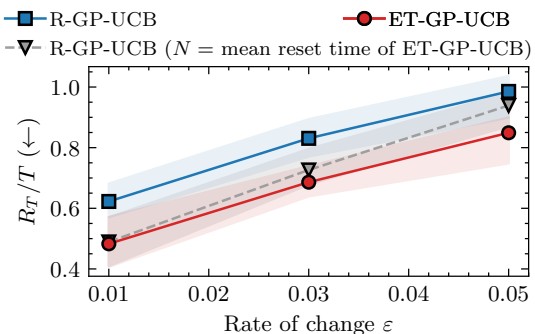

Table 2: Online MLE hyperparameter optimization of the GP when using ET-GP-UCB.

| Algorithm | $\varepsilon = 0.01$ $R_T/T$ ($\downarrow$) | $\varepsilon = 0.03$ $R_T/T$ ($\downarrow$) | $\varepsilon = 0.05$ $R_T/T$ ($\downarrow$) |
|---|---|---|---|
| GP-UCB | 0.835 | 1.260 | 1.405 |
| R-GP-UCB | 0.776 | 0.998 | 1.123 |
| ET-GP-UCB* | 0.612 | 0.870 | 1.057 |

*with "learn-then-monitor" approach

Figure 4: Impact of Reset Frequency vs. Timing.

The results over 20 runs are displayed in Figure 3 (d). We can observe that ET-GP-UCB is the best performing algorithm since it can detect the change in the objective function and converge to the new optimal solution. We can further see that TV-GP-UCB and R-GP-UCB, which rely on an ongoing and smooth change in the objective function, empirically struggle to cope with such sudden changes resulting in significantly increased regret. While UI-TVBO can also recover from the change, ET-GP-UCB still outperforms this baseline. The example also highlights that a constant reset time that is too short can yield divergent behavior in higher-dimensional settings and non-smooth changes.

### 6.2.4 Ablation on the Impact of Reset Frequency vs. Reset Timing

To better understand the source of empirical improvement of ET-GP-UCB, we conducted an ablation study to test whether the performance gains are due to resetting at the correct frequency or at the correct time steps. Specifically, we investigated the effect of using the average resetting time of ET-GP-UCB, derived from Monte-Carlo simulations of the event trigger, as the reset interval $N$ for R-GP-UCB. Figure 4 shows within-model experiments across multiple rate of changes for 50 different seeds. The results demonstrate that resetting at the average frequency of the trigger improves regret performance at low rates of change. However, as the rate of change increases, adaptively resetting at the correct time steps, i.e., considering the realized changes in the objective function, becomes crucial for minimizing regret.

### 6.3 Practical Extensions of ET-GP-UCB

Next, we discuss practical extensions to ET-GP-UCB that can enhance its performance in practice.

**Upper Bound on the Noise** Imposing a fixed upper bound on the noise term $\bar{w}_{t_r}$ after a certain number of time steps can be practical. While the gradual increase of $\bar{w}_{t_r}$ is crucial for Lemma 3 to hold, it results in large thresholds in (11) over time. This might lead to undetected changes, especially when combined with a large $\bar{N}$ and rare changes in the objective. Considering bounded noise in Assumption 1 instead of Gaussian noise might also be more suitable for some applications.

**Online Hyperparameter Optimization** Practitioners should be cautious when using naïve MLE for all hyperparameters, as our algorithm fits a *time-invariant* GP to *time-varying* data. This can inflate the noise variance, again resulting in large thresholds in (11). The core issue lies in the dual use of the model mismatch: the event-trigger mechanism is designed to detect model mismatches, while hyperparameter optimization aims to minimize this mismatch. These two objectives conflict. We suggest estimating the noise on hold-out data, as demonstrated in Section 6.2 with the temperature dataset. If this is not feasible, we recommend using reasonable priors and constraints for all hyperparameters and then performing maximum-a-posteriori estimation, as is common practice for most GP and BO libraries (GPy, 2012; Gardner et al., 2018; Balandat et al., 2020) and demonstrated in the policy search example. In the case of no prior knowledge, a simple "learn-then-monitor" approach, i.e., learning the parameters and freezing them afterward while monitoring for changes, can be helpful. To demonstrate this, we conducted experiments with online hyperparameter optimization for all baselines using only very coarse constraints. Specifically, we use a lower bound of 0.001 and an upper bound of 0.1 for the noise (true noise value is 0.02) and a lower bound of 0.01 and an upper bound of 1 for the lengthscales (true lengthscales are 0.2). For ET-GP-UCB, we use the first $2 \cdot d$ time steps

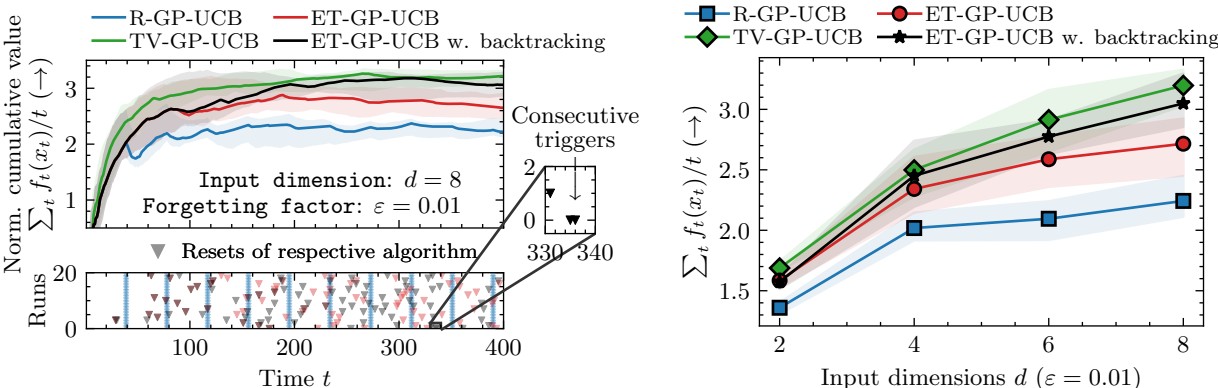

Figure 5: *Left:* Performance on an eight-dimensional within-model objective function. Retaining a portion of past data using Algorithm 2 leads to a significant performance boost, nearly matching the results of TV-GP-UCB, even without prior knowledge of $\varepsilon$. *Right:* Final performance for different dimensions ($d \in \{2, 4, 6, 8\}$) with a fixed $\varepsilon$. As dimensionality increases, the benefit of retaining data becomes more pronounced.

after a trigger event to learn the noise variance and individual lengthscales, and then freeze the parameters and start monitoring for changes using our event trigger. Table 2 shows that this approach decouples the two conflicting objectives, yielding reasonable performance without prior knowledge of $\varepsilon$, the spatial hyperparameters, nor the observation model.

**Retaining Past Information**   Exploring alternative strategies to augment the data set instead of performing a full reset every time the event trigger activates is interesting to preserve the most recent information. In this paper, we demonstrated that even with a full reset we obtain superior performance on commonly used real-world benchmarks but preserving information can become crucial for higher-dimensional problems. One idea that arises directly from our event-triggered setup is to implement a "backtracking" mechanism when a change is detected: after detecting a change, the algorithm backtracks to check how many of the most recent data points still satisfy the trigger bounds. These relevant observations are then retained for further optimization. By re-using the event trigger to determine which data to keep, we ensure that, in the event of sudden changes, there are no inconsistent data points in the data set. We cap the re-initialization size of the data set by $2 \cdot d$; similar data set sizes are commonly used to initialize BO algorithms (Eriksson et al., 2019; Balandat et al., 2020; Müller et al., 2021). The backtracking procedure is summarized in Algorithm 2.

To test this approach, we conduct within-model comparisons for different dimensions summarized in Figure 5. Retaining some of the past data proves to be very beneficial. In Figure 5 (left), we can see that by using Algorithm 2, ET-GP-UCB is almost able to match the performance of TV-GP-UCB even without prior knowledge of $\varepsilon$. Notably, combining data reuse and smooth changes can yield consecutive resets, occasionally transforming the method into an adaptive sliding window approach. From a computational standpoint, this poses no significant overhead, as backtracking through the dataset and evaluating the event trigger are computationally efficient.

---

**Algorithm 2** Backtracking procedure to retain data

1: **procedure** BACKTRACKING($\mathcal{D}$, EVENT TRIGGER)
2:    $\hat{\mathcal{D}}_{\mathrm{keep}} \leftarrow \{\}$          ▷ Initialize an empty data set
3:    **for** $i \leftarrow |\mathcal{D}| - 1$ **downto** 0 **do**
4:       $(y_i, \boldsymbol{x}_i) \leftarrow \mathcal{D}[i]$     ▷ Extract input-output pair
5:       $\gamma_{\mathrm{reset}} \leftarrow$ EVENT TRIGGER($\hat{\mathcal{D}}_{\mathrm{keep}}$, $(y_i, \boldsymbol{x}_i)$)
6:       **if** $\gamma_{\mathrm{reset}}$ **or** $|\hat{\mathcal{D}}_{\mathrm{keep}}| \geq 2 \cdot d$ **then**
7:          **break**                    ▷ Stop backtracking
8:       $\hat{\mathcal{D}}_{\mathrm{keep}} \leftarrow \hat{\mathcal{D}}_{\mathrm{keep}} \cup \{(y_i, \boldsymbol{x}_i)\}$
9:    **return** $\hat{\mathcal{D}}_{\mathrm{keep}}$

---

Figure 5 (right) shows that as the dimensions increase, using a strategy that reuses some of the past data becomes more beneficial. The results for more forgetting factors are in Appendix A.

# 7   Conclusion and Future Work

This paper presented the novel concept of event-triggered resets to optimize time-varying objective functions with TVBO. Our theoretical results show that algorithms that reset between lower and upper thresholds yield meaningful regret even if the rate of change is unknown. This allowed us to optimize the actual reset timing for increased empirical performance using probabilistic error bounds. The resulting algorithm, ET-GP-UCB, adapts to the realized changes in the objective function without relying on exact prior knowledge of these temporal changes, making it especially useful in real-world scenarios. We empirically showed that the adaptive resetting of ET-GP-UCB consistently outperforms periodic resetting and further demonstrated ET-GP-UCB's increased performance on real-world benchmarks. Lastly, we presented practical extensions to our algorithm. These extensions included an approach for practical online hyperparameter estimation within the event trigger framework and an approach to retaining past information, which further increases performance in higher-dimensional settings. The latter leverages the event trigger not only to detect changes but also to determine how much recent data can be reused, thereby improving sample efficiency. Overall, ET-GP-UCB offers an adaptive solution for optimizing time-varying objectives with minimal hyperparameter tuning and prior knowledge required, making it a valuable tool for many time-varying optimization problems.

An interesting direction for future work on ET-GP-UCB is its further theoretical analysis. While Corollary 1 ensures the desirable scaling of the regret for worst-case resets, it does not reflect the improved empirical regret when increasing the window size as the event trigger is not explicitly included in the analysis. Regret bounds that explicitly consider the choice of event trigger are a promising direction for future research and would bridge the current gap between theoretical analysis and empirical performance. For further discussion, see Appendix C. Another research direction would be to move away from a setting solely focused on dynamic regret and consider e.g., a setting with multiple comparators that are fixed for certain window lengths (Zhang et al., 2020). Here, we believe that adaptive approaches as proposed herein can be especially beneficial. Lastly, investigating different event triggers within the framework of Definition 2 is interesting, such as event triggers that consider multiple time steps in their test and threshold function to, e.g., identify slow trends in the objective. New and tailored data augmentation strategies may also arise with new event triggers or event triggers tailored to specific applications.

## Acknowledgments

The authors thank Pierre-François Massiani, Dominik Baumann, Christian Fiedler, and Noel Brindise for their helpful comments and discussions, as well as the reviewers and action editor for their constructive feedback, which helped improve the final version of the paper. This work is partially funded by the Deutsche Forschungsgemeinschaft (DFG, German Research Foundation)–RTG 2236/2 (UnRAVeL). Simulations were performed with computing resources granted by RWTH Aachen University under project rwth1579.

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

# Supplementary Material for "Event-Triggered Time-Varying Bayesian Optimization"

Following is the technical appendix. Note that all citations here are in the bibliography of the main document, and similarly for many of the cross-references.

## A   Additional Results and Details on Experimental Design

This section contains details about the design of the experiment for the results in Section 6.2 for reproducibility.

### A.1   Creating Within-Model Objective Functions

To create the within-model objective functions, we take two different approaches. For the two-dimensional objectives in Section 6.2.1, we draw sample paths $g_i$ from a GP with the corresponding hyperparameters on a dense equidistant grid of the compact set $\mathbb{X} = [0,1]^2$ with $100 \times 100$ input locations. To create the time-varying objective function $f_t$ we then combine the sample paths following Assumption 2 with a specified rate of change $\varepsilon$ and perform linear interpolation in the dense grid. For the higher-dimensional within-model objective functions in Section 6.3, this procedure is not feasible. Therefore, we approximate prior samples $g_i$ with a $M$ random Fourier features (RFFs) following Rahimi & Recht (2007). This yields a set of parametric functions

$$g_i(\boldsymbol{x}) = \sum_{m=1}^{M} w_m \phi_m(\boldsymbol{x}) \quad \text{with} \quad \phi_m(\boldsymbol{x}) = \sqrt{\frac{2}{M}} \cos(\boldsymbol{\theta}_m^\top \boldsymbol{x} + \tau_m). \tag{12}$$

Here, $\boldsymbol{\theta}_m$ are sampled proportional to the kernel's spectral density and $\tau_m \sim \mathcal{U}(0, 2\pi)$. These approximate sample paths again can be combined following Assumption 2 to yield the objective function $f_t(\boldsymbol{x})$ given $\varepsilon$. In all experiments in Section 6.3, we use $M = 1028$ RFFs. This approach does not require interpolation in the input space since the approximate sample paths are parametric.

### A.2   Additional Results on Within-Model Objective Functions

Here, we list some of the additional results on within-model objective functions. On the right in Figure 6, we conducted an experiment with a change in the rate of change after 175 time steps (segment highlighted in gray). We initialize all algorithms that require epsilon as an input to their algorithm with the rate of change at time step 0. In this experiment, the adaptive nature of our algorithm becomes evident. After the change in the rate of change $\varepsilon$, ET-GP-UCB starts to increase the reset frequency adjusting to the higher rate of change in the unknown objective. We can observe how TV-GP-UCB with the too small rate of change starts to diverge over time. While the difference for 500 time steps is not as big, there is a clear diverging trend.

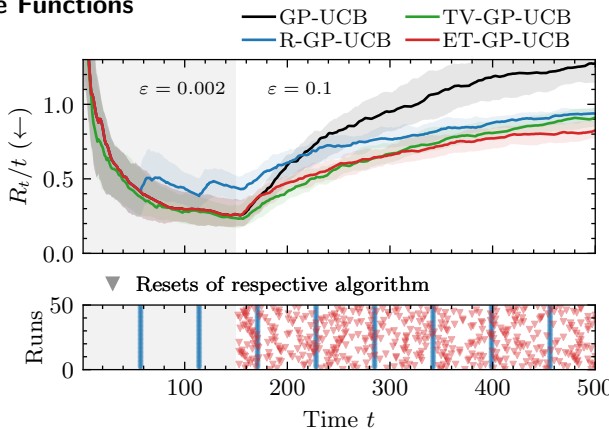

Figure 6: Within-model with change in $\varepsilon$.

We next further visualize the results from Table 1 in Figure 7. Here, we only list the best-performing variant of ET-GP-UCB to increase the readability of the plot. Lastly, Figure 8 shows the results of using Algorithm 2 to keep some of the past information for different rate of changes. We can observe that using the backtracking procedure improves the performance for all dimensions for the given rate of changes.

### A.3   Comparison to Performance-based Triggers

Another idea that arises naturally for an event trigger is to consider a reset strategy that is solely based on an improvement in performance. This strategy is inspired by trust-region BO, TuRBO (Eriksson et al., 2019), where a lack of improvement "triggers" a refinement of the search domain. We apply a variant of this mechanism to the time-varying setting: If an algorithm is not able to improve its performance for $\tau_{\text{fail}} \in \mathbb{N}_+$ time steps in a row, the event trigger resets the data set. To set this threshold, we use the same heuristic proposed in the TuRBO paper and set $\tau_{\text{fail}} = \lceil D/q \rceil = D$ (c.f. Appendix D in Eriksson et al. (2019) and our batch size is $q = 1$ given our problem setting). We refer to this event trigger as TuRBO Trigger. To test its performance, we ran the same benchmark from the previous section, i.e., multiple within-model functions generated using RFFs (cf. Section A.1) with different rates of change and different dimensions. The results are in Figure 9.

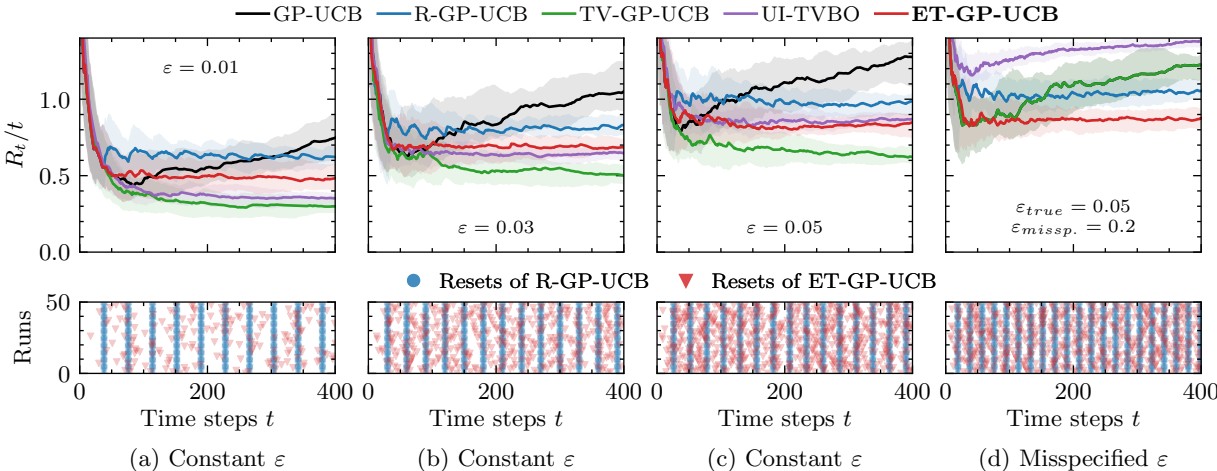

Figure 7: *Top row:* performance of the different algorithms in terms of normalized regret. The shaded areas show one standard deviation of the respective algorithms. *Bottom row:* reset timings of R-GP-UCB and ET-GP-UCB. In (a) to (c), rate of change is fixed for all time steps, and in (d), it is misspecified for TV-GP-UCB, UI-TVBO, and R-GP-UCB. In (a) to (c), TV-GP-UCB is the full information reference, showing the best possible UCB performance with known constant $\varepsilon$. We compare ET-GP-UCB to R-GP-UCB, the latter of which uses the true $\varepsilon$ to determine its periodic reset time. For ET-GP-UCB, we use the same parameter $\delta_{\mathrm{B}}$ in the event trigger for the test cases (a) to (d). In all cases, ET-GP-UCB outperforms R-GP-UCB. It also outperforms TV-GP-UCB and UI-TVBO if $\varepsilon$ is misspecified, as shown in (d), since ET-GP-UCB does not rely on prior information on $\varepsilon$. The plots show the median performance and the interquartile.

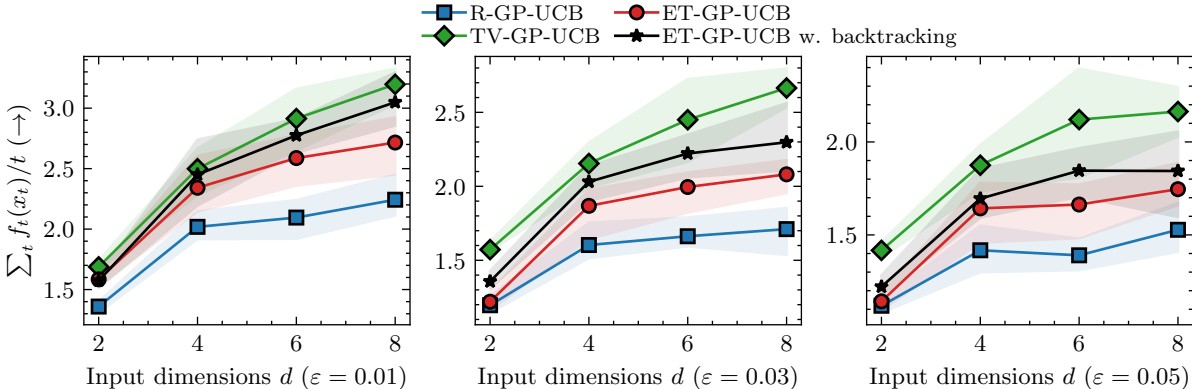

Figure 8: Benefit of reusing some of the past information through Algorithm 2 for different $\varepsilon$.

We can observe that for all rates of change, the TuRBO trigger performs worse than the other baselines. Unlike in TuRBO, a trigger event significantly impacts performance since all data is discarded (while shrinking the trust region for TuRBO is less drastic). This is especially the case for lower-dimensional objective functions where we only allow for a few failures before the dataset is reset. An additional factor is that the objective function is noisy. Therefore, even consecutive evaluations at the optimum of a non-changing function could yield a reset, which, in contrast to TuRBO, requires significant new data to recover from. While performance-based resets hold promise as they are not as tightly coupled to the prior assumptions of the GP, they require further in-depth investigation and algorithm design for the dynamic settings to yield well-performing approaches by further including heuristics. Also, further tuning of $\tau_{\mathrm{fail}}$ for each problem could improve its performance.

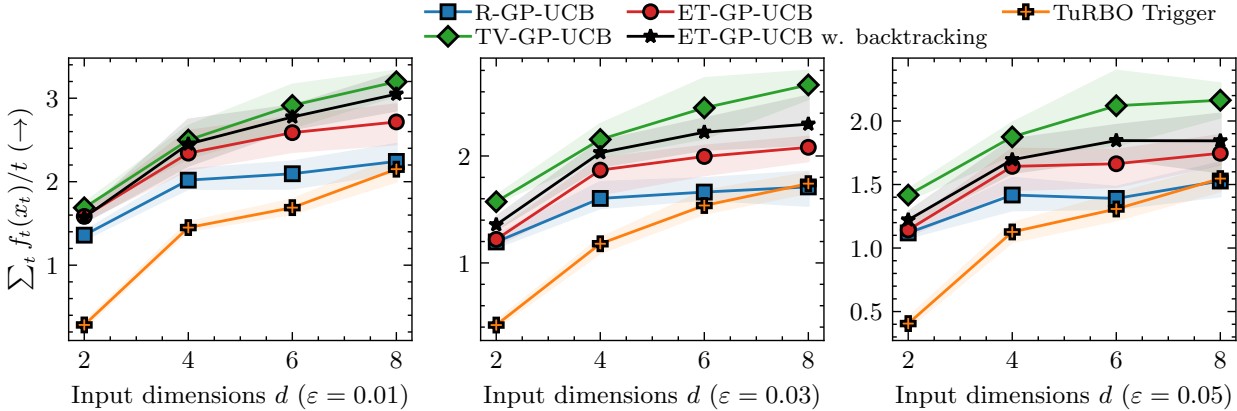

Figure 9: Comparison to a performance-based event trigger (TuRBO Trigger).

## A.4 Performance on Standard Optimization Benchmarks

Next, we will test our event trigger on some standard benchmarks, specifically Ackley ($d = 2$) and Hartmann ($d = 6$), which are popular for testing BO algorithms. The event trigger in ET-GP-UCB is tightly coupled to the GP prior. We, therefore, expect some false positives of our event trigger on these benchmarks. Similar to Section 6.2.2 and Section 6.2.3, the comparisons in this section aim to test to what extent resets affect the empirical performance. For Ackley, we set the time horizon to $T = 60$ and induce a shift in the optimum after $T/2$ time steps. Similarly, we set Hartmann's time horizon to $T = 200$ and induce the shift after $T/2$ time steps. To induce this shift, we perturb the input of the objective function by a constant factor.

We use the same UCB acquisition function stated in Section 6, run ten different seeds, and learn the lengthscales $\ell_i$ online for all baselines. For the lengthscales, we use corse bounds as $\ell_i \in [0.005, 4.]$ (Eriksson et al., 2019) and standardize the output data at each time step given the current data set which follows best practices in BO (Balandat et al., 2020). Since we learn the lengthscales online, we apply the "learn-then-monitor" approach introduced Section 6.3 for the ET-GP-UCB baselines and use the same $\delta_B$ as in all previous experiments. For TV-GP-UCB, we set the rate of change to $\varepsilon = 0.01$ to yield good performance in the static setting but still allow for some adaptation.

Figure 10 shows the performance of all baselines. For Ackley, we can observe that at the beginning of optimization, there are some resets of both ET-GP-UCB variants. However, we can also observe that these resets do not significantly impact the final performance (before the change) when considering simple regret. After the shift, we can observe that all the time-varying baselines can adjust to the change in the objective, and as expected, GP-UCB struggles to cope with the stale data in the data set.

On Hartmann, the event-triggered baseline without backtracking struggles and is outperformed by GP-UCB. Similar to Ackley, we can observe early resets of both ET-GP-UCB variants. In this higher-dimensional example, these resets are more costly for vanilla ET-GP-UCB. For the backtracking variant, we can observe consecutive resets resulting in an event-triggered sliding window (cf. Section 6.3). With this backtracking mechanism, the ET-GP-UCB variant outperforms GP-UCB significantly in terms of simple and cumulative regret. Similar to the Ackley example, most of the time-varying baselines can better adjust to the shift in the optimum compared to GP-UCB. Noteworthy is also that ET-GP-UCB with backtracking significantly outperforms GP-UCB in terms of simple regret even before the change. Removing some initial data allows this variant to achieve better GP hyperparameters and converge to the optimum. This result raises further questions on how classic BO algorithms should deal with past data for better convergence.

In summary, we observed that ET-GP-UCB does yield additional resets when optimizing an objective function that is not well explained by the GP prior, but the impact on final performance is relatively small for the considered test functions when leveraging the proposed backtracking mechanism. We further observed that this mismatch in prior and objective functions can also harm static algorithms. Additionally, we observe

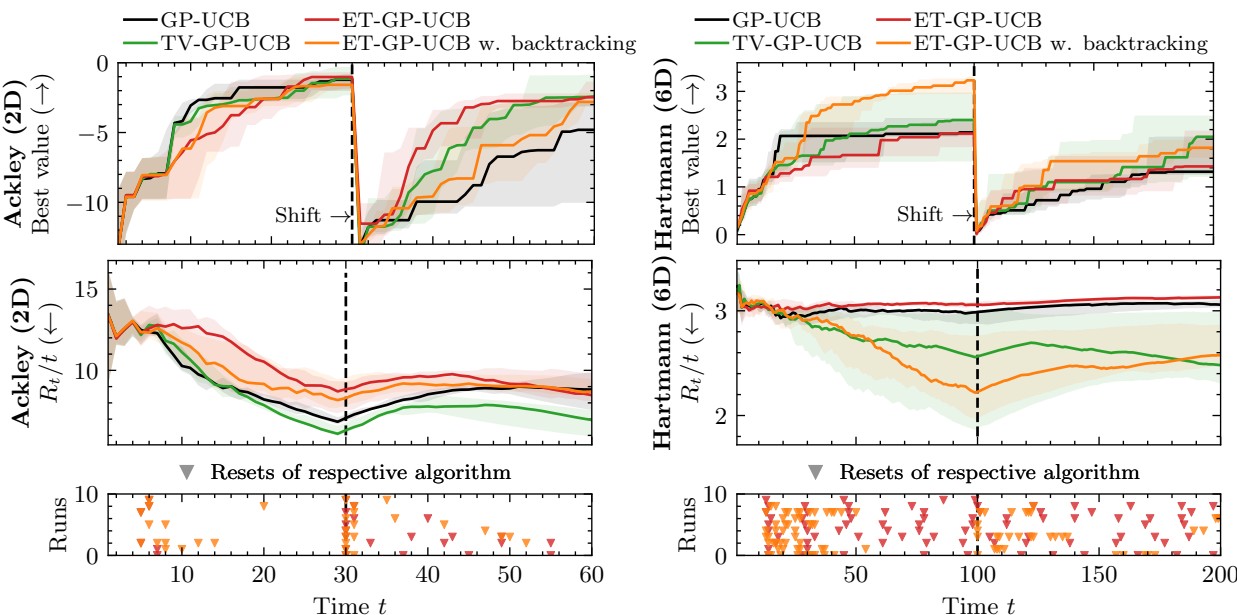

Figure 10: Comparison on the standard benchmarks Ackley (left) and Hartmann (right). We show the best obtained value on top and the normalized cumulative regret on the bottom. We reset the best value an algorithm as observed after the shift (indicated by the dashed line) to test how the algorithms cope with the shift in the optimum. All plots show the median performance as well as the interquantiles.

that the same algorithms developed to tackle time-varying problems may also be helpful for static problems to overcome some problems with past data points, which is an exciting research direction.

### A.5 Comparisons to Algorithms in the Finite Variational Budget Setting

In this section of the appendix, we compare ET-GP-UCB to W-GP-UCB (Deng et al., 2022), and BR-GP-UCB and SW-GP-UCB (Zhou & Shroff, 2021). As discussed in Section 3, their underlying assumptions differ from ours in that they assume a fixed variational budget. Therefore, a theoretical comparison is not reasonable, but an empirical comparison, especially on real-world data, is interesting. As the baseline that does not consider time-variations, we also compare against IGP-UCB (Chowdhury & Gopalan, 2017), which is an improved form of the agnostic GP-UCB of Srinivas et al. (2010). For the comparison, we use the code base submitted by Deng et al. (2022) in the supplementary material of https://proceedings.mlr.press/v151/deng22b. Their experiments consist of:

(a) an one-dimensional experiment with sudden changes at time steps $t = 100$ and $t = 200$,
(b) an one-dimensional experiment with a slowly changing objective function,
(c) an experiment on real-world stock market data.[4]

We refer to Deng et al. (2022) for the details on how the objective function for (a) and (b) is composed. We implemented our algorithm ET-GP-UCB in their MATLAB code base and added random search as a baseline.

While going through the implementation details, we noted that in the stock market data benchmark, the data was not normalized and values of the objective function ranged from 19 to 250. This is problematic as all algorithms use a prior mean of zero and a prior variance of one. Thus, all observations will have a very small likelihood under the given model. Secondly, the noise variance was set to 300, which seems high for stock data where observation noise should be low. The high noise variance combined with the prior signal variance of one also explains the very high standard deviations of all the algorithms in Deng et al. (2022,

---

[4]Note that stock market data does not fit the fixed variational budget setting as stock prices constantly change. However, since this example was used in prior work, we treat it as a benchmark but wanted to highlight that *ongoing change* is the more suitable assumption.

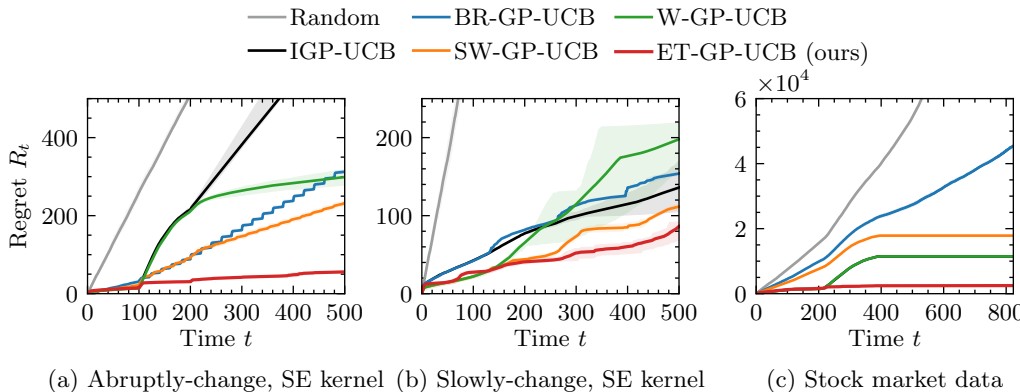

(a) Abruptly-change, SE kernel  (b) Slowly-change, SE kernel  (c) Stock market data

Figure 11: Performance using the refactored GP prior in the experiments of Deng et al. (2022).

Figure 1 (c)). Overall, the stock market data has a very low marginal likelihood for the model chosen in Deng et al. (2022). Therefore, we decided to improve the model and add a preprocessing step to the stock market data example by normalizing the data to a mean of zero and a variance of one to fit the prior model assumptions of all algorithms. Next, we chose a noise variance of 0.01 to account for smaller short-term fluctuations in stock prices. We also changed the noise variance of the synthetic experiments (a) and (b) to 0.1. It was previously set to 1 (same as the signal variance), amounting to a 'signal to noise ratio' of 1, which is very low for practical optimization problems. We then re-ran their experiments using 20 runs for the synthetic experiments and 5 runs for the stock market data as in Deng et al. (2022). For ET-GP-UCB, we use the same parameters for the event trigger as in all previous experiments, as well as no hard upper and lower bounds, as this resulted in the best empirical performance in Section 6.2. Note that all other time-varying algorithms rely on specifying the amount of change $B_T$ a priori, which is hard to estimate in practice. The results are shown in Figure 11 and regret at the last time step is listed in Table 3.

We can see that ET-GP-UCB can outperform all baselines in experiments Figure 11 (a) to (c). Especially in Figure 11 (a), ET-GP-UCB can detect the sudden changes in the objective function and then quickly find the new optimum due to the simple one-dimensional objective function. In Figure 11 (b), ET-GP-UCB is the best performing algorithm, but W-GP-UCB is competitive. Considering the good performance of IGP-UCB, we can also conclude that the changes in the objective function over the time horizon are relatively small. In Figure 11 (c), ET-GP-UCB is again the best-performing algorithm. We also observe that in Figure 11 (c) *all* algorithms significantly improve in performance and yield a smaller standard deviation compared to Deng et al. (2022, Figure 1 (c)) because the GP prior model is more suitable for modeling the objective function. After approx. 500 time steps ET-GP-UCB, W-GP-UCB, SW-GP-UCB, and IGP-UCB (see Table 3), no longer accumulate regret. Looking at the objective function, this is expected as the optimal bandit arm no longer changes and the optimal values do not change significantly.

Table 3: Comparison on experiments in Deng et al. (2022).

| Algorithm | (a) Abruptly-change Regret ($\mu \pm \sigma$) | (b) Slowly-change Regret ($\mu \pm \sigma$) | (c) Stock market Regret ($\mu \pm \sigma$) |
|---|---|---|---|
| IGP-UCB | 704.6±238.9 | 136.0±32.2 | 11 462.3±56.0 |
| BR-GP-UCB | 312.5±4.3 | 153.9±11.0 | 45 362.2±430.1 |
| SW-GP-UCB | 233.2±9.1 | 111.6±15.7 | 17 854.9±27.9 |
| W-GP-UCB | 298.8±27.3 | 197.0±29.7 | 11 461.3±32.9 |
| ET-GP-UCB (ours) | **55.8**±2.7 | **86.3**±42.8 | **2 483.8**±39.8 |

**Results with the Original Gaussian Process Prior**

Below in Figure 12 are the results using the original GP prior from Deng et al. (2022). We can observe that for Figure 12 (a), ET-GP-UCB can reliably detect the first change in the objective function at $t = 100$. However, due to the low signal-to-noise ratio, ET-GP-UCB can only detect the second change approximately 50% of the time. It still outperforms periodic resetting and the sliding window approach. In Figure 12 (b), ET-GP-UCB shows a similar performance as W-GP-UCB. Both examples, however, highlight that ET-GP-UCB is not as dominant as in previous examples if the signal-to-noise ratio is too low since $\bar{w}_t$ will be large relative to $|y_t - \mu_{\mathcal{D}_t}(\boldsymbol{x}_t)|$. For the stock market data, we observe the discussed large standard deviations. All algorithms outperform the random baseline. However, we can observe that the ordering of algorithms has changed compared to Deng et al. (2022, Figure 1 (c)) even though we used their implementations and only added ET-GP-UCB and the random baseline to the run script. Further, changing the random seed again changed this ordering of the algorithms, which indicates that the prior model assumptions yield random-like behavior as the prior does not allow the GP model to make meaningful decisions.

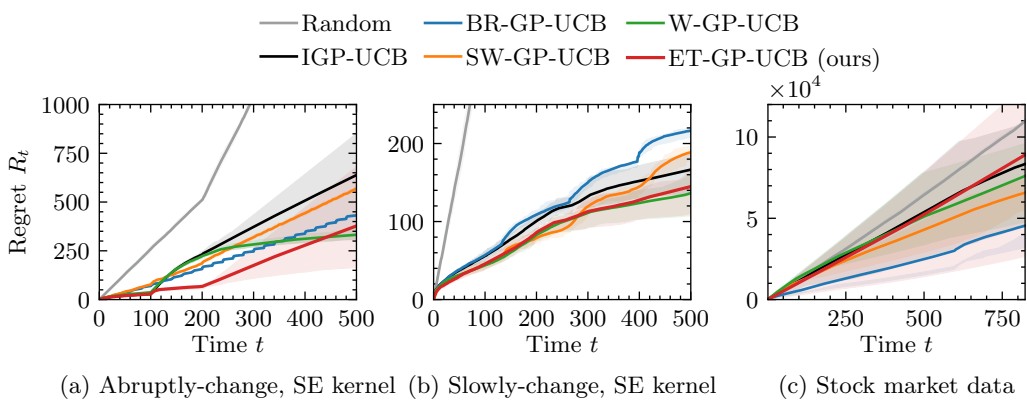

(a) Abruptly-change, SE kernel    (b) Slowly-change, SE kernel    (c) Stock market data

Figure 12: Performance using the original GP prior for the experiments of Deng et al. (2022).

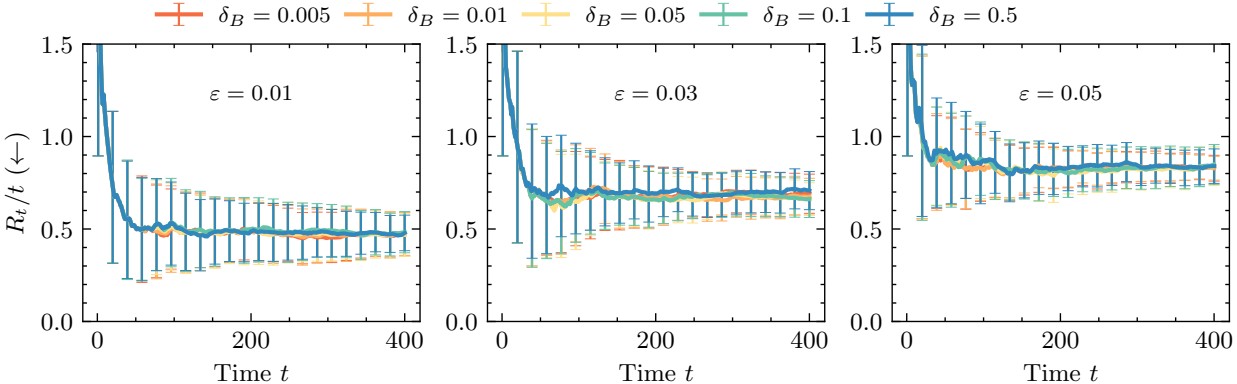

Figure 13: Sensitivity of our event trigger from (11) with respect to its hyperparameter $\delta_{\mathrm{B}}$.

## B Sensitivity of ET-GP-UCB to the Event Trigger Hyperparameter $\delta_{\mathrm{B}}$

Table 4 and Figure 13 show the sensitivity of ET-GP-UCB to the event trigger parameter $\delta_{\mathrm{B}}$ in Lemma 3. To solely focus on the influence of $\delta_{\mathrm{B}}$ with use the maximum window size as $W = T$. It should be noted, that for different window sizes, different $\delta_{\mathrm{B}}$ may be optimal for the best possible empirical performance as both, the location of lower and upper reset bounds, and the hyperparameter $\delta_{\mathrm{B}}$, influence how much (and where) probability mass from the distribution over reset times is cut off. For each $\delta_{\mathrm{B}}$, we ran 50 different seeds on within-model objective functions (cf. Section A.1). All hyperparameters are listed in Table 12. As $\delta_{\mathrm{B}}$ is increased, the average number of resets increases as the probabilistic uniform error bound is softened. We observe that for different rates of changes $\varepsilon$ different $\delta_{\mathrm{B}}$ may be optimal to obtain the lowest possible empirical regret. From Figure 13, we can observe that for higher rates of change a smaller $\delta_{\mathrm{B}}$ can yield a smaller variance in the final regret. In summary, ET-GP-UCB displays a low sensitivity to $\delta_{\mathrm{B}}$ highlighting that our algorithm is readily applicable without the need for excessive hyperparameter tuning. In practice, we found $\delta_{\mathrm{B}} = 0.1$ to yield good empirical performance in all of our experiments.

Table 4: Sensitivity of ET-GP-UCB to $\delta_{\mathrm{B}}$ in Lemma 3.

| Rate of Change $\varepsilon$ | ET-GP-UCB Hyperparameter $\delta_{\mathrm{B}}$ | Avg. No. of Resets | Avg. Regret $R_T$ |
|---|---|---|---|
| | 0.005 | 2.66 | 191.39 |
| | 0.01 | 2.96 | 193.05 |
| $\varepsilon = 0.01$ | 0.05 | 3.20 | 200.16 |
| | 0.1 | 3.38 | 200.33 |
| | 0.5 | 3.98 | 196.03 |
| | 0.005 | 6.42 | 276.84 |
| | 0.01 | 6.82 | 273.37 |
| $\varepsilon = 0.03$ | 0.05 | 7.72 | 269.01 |
| | 0.1 | 8.04 | 271.59 |
| | 0.5 | 10.32 | 280.05 |
| | 0.005 | 9.60 | 331.69 |
| | 0.01 | 9.96 | 328.74 |
| $\varepsilon = 0.05$ | 0.05 | 11.40 | 329.47 |
| | 0.1 | 11.88 | 332.04 |
| | 0.5 | 14.44 | 334.80 |

## C  Discussion on the Theoretical Analysis and Future Directions

The current theoretical analysis does not explicitly include the event trigger (11). As stated in Section 7, we are interested in tighter regret bounds, tailored to the specific event trigger. This has the potential for additional insights into the design of new triggers. In the following, we list some of the challenges that may arise when conducting this theoretical investigation and give some pointers that may be helpful to derive tighter bounds.

**Reset times are stopping times.** First, note that the activation time of the event trigger is a random variable as $\tau := \min\{t \in \mathbb{N} : \psi_t(\boldsymbol{x}_t) > \kappa_t(\boldsymbol{x}_t)\}$. This random variable $\tau$ is a stopping time, as $\psi_t(\boldsymbol{x}_t)$ is a discrete-time stochastic process and the trigger event is completely determined by the information known up to $t$. The stochastic process depends recursively on the acquisition function, which makes an analytical study of the stopping time difficult; even for simple stochastic processes, calculating expected values of stopping times is typically intractable.

**The regret analysis should be specific for the event trigger.** There is some information available on $\tau$: As long as $T$ is finite, $\tau$ is a bounded random variable as $\tau \in [1, T]$. This also means, that $\tau$ is a sub-Gaussian random variable. One can use concentration inequalities of sub-Gaussian random variables to bound the maximum of $\tau$ with high probability and then use this bound in the theoretical analysis. For the analysis to be specific for the event trigger, this concentration inequality has to depend on the test and threshold functions of the event trigger which, as stated above, is in itself very challenging: Random sampling block lengths within $[1, T]$ have the same sub-Gaussian property, i.e., with the same sub-Gaussian constant. However, the empirical performance would be very different compared to a sophisticated event trigger. Using standard bounds for sub-Gaussian random variables would therefore likely still result in a mismatch between empirical performance and theoretical analysis.

**Bounding the random block sizes and number of block simultaneously is promising.** Once a *event trigger specific* bound on the stopping time is established, the theoretical analysis would still require a fundamentally new way of quantifying the model mismatch compared to Theorem 1. The current analysis bounds the maximum model mismatch by considering the maximum block length at every time step. Here, one could substitute in the bound on the stopping time $\tau$ (which depends on $T$). However, the bound only holds with high probability and, therefore, for a valid probabilistic statement, a union bound over all time steps is required. This union bound can result in a super-linear scaling of the regret bound. An interesting direction for future work would be instead of considering the maximum model mismatch at each time step to bound the composition of the number of blocks and size of the blocks simultaneously.

## D   Influence of the window size on the distribution of resets for different $\varepsilon$

In Figure 2 (left), we displayed the influence of the window size on the distribution of reset times. In Figure 14, we show this influence also for different $\varepsilon$. We further add the empirical cumulative distribution in the right column.

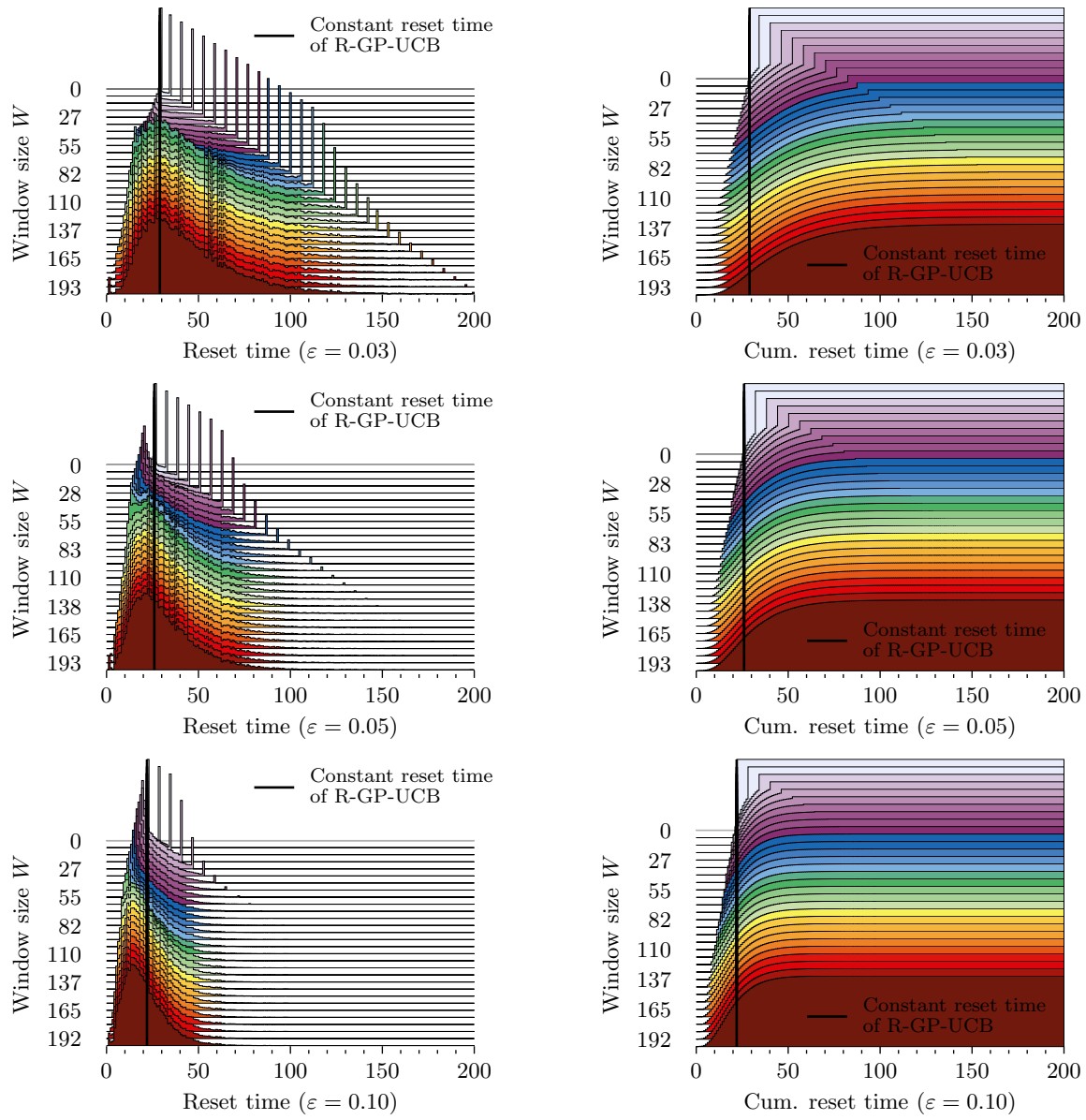

Figure 14: *Left column:* Influence of the window size on the empirical PDF of reset times. *Right column:* Influence of the window size on the empirical CDF.

## E   Overview of the Temperature Data

In Figure 15, we show the normalized temperature data used for the empirical kernel in the results in Figure 3. The full pre-processing of the temperature data is described in the corresponding jupyter-notebook in the provided code base.

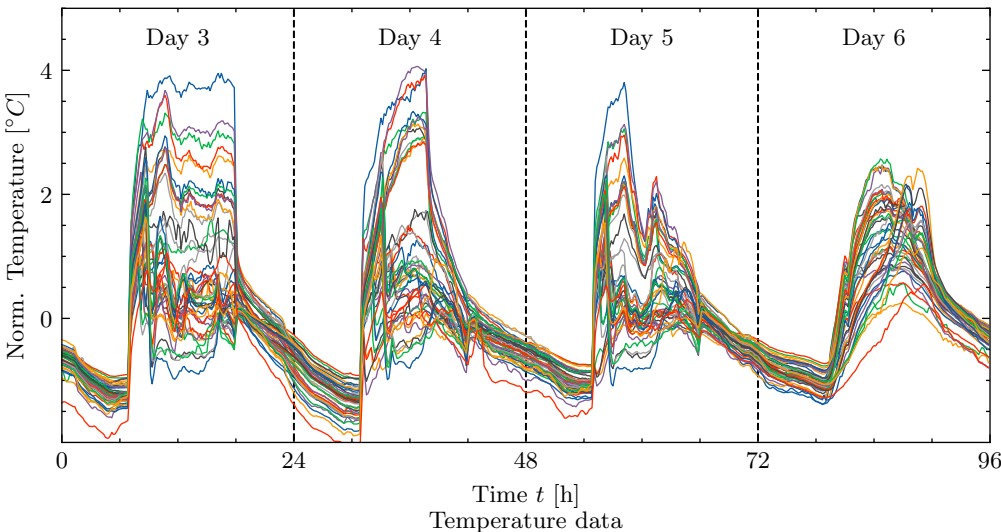

Figure 15: Normalized temperature data used for the results in Figure 3. Each color resembles the measurement data from one sensor.

## F   Proof of Theorem 1 and Corollary 1 from Section 4

### F.1   Proof of Theorem 1

**Restated Theorem 1**   *Let the domain* $\mathbb{X} \subset [0, r]^d \subset \mathbb{R}^d$ *be convex and compact with* $d \in \mathbb{N}_+$ *and let* $f_t$ *follow Assumption 2 with a kernel* $k(\boldsymbol{x}, \boldsymbol{x}')$ *such that Assumption 3 is satisfied. Pick* $\delta \in (0, 1)$ *and set* $\beta_t = 2 \ln \left( 2\pi^2 t^2 / (3\delta) \right) + 2d \ln \left( t^2 db_1 r \sqrt{\ln(2da_1\pi^2 t^2/(3\delta))} \right)$. *Pick a lower block length* $\underline{N} \in \mathbb{N}_+$ *and a upper block length* $\bar{N} \in \mathbb{N}_+$ *with* $\underline{N} \leq \bar{N} \leq T$. *Let* $\gamma_{\bar{N}}$ *be the maximum information gain over* $\bar{N}$ *points. Then, running any algorithm with block sizes between* $\underline{N}$ *and* $\bar{N}$ *satisfies*

$$R_T \leq \sqrt{C_1 T \beta_T \left( \frac{T}{\underline{N}} + 1 \right) \gamma_{\bar{N}} + 2} + T\phi_T(\varepsilon, \bar{N}) \tag{13}$$

*with probability at least* $1 - \delta$, *where* $C_1 = 8/\ln(1 + \sigma_\mathrm{n}^{-2})$ *and we defined*

$$\phi_T(\varepsilon, \bar{N}) := 2\sqrt{\beta_T(3\sigma_\mathrm{n}^{-2} + \sigma_\mathrm{n}^{-4})\bar{N}^3 \varepsilon} + 2(\sigma_\mathrm{n}^{-2} + \sigma_\mathrm{n}^{-4})\bar{N}^3 \varepsilon(b_0 + \sqrt{2}\sigma_\mathrm{n})\sqrt{\ln \frac{4(a_0+1)\pi^2 T^2}{3\delta}}. \tag{14}$$

**Proof:**   The proof of Theorem 1 is mainly based on the proof of Bogunovic et al. (2016, Theorem 4.2) which builds on arguments from Srinivas et al. (2010). The key distinction of our analysis is that we will consider the block length not to be fixed but within an interval $[\underline{N}, \bar{N}]$. For completeness, we list the full proof here. As in Bogunovic et al. (2016) and Srinivas et al. (2010), we first fix a discretization $\mathbb{X}_t \subset \mathbb{X} \subseteq [0, r]^d$ of size $(\vartheta_t)^d$ satisfying

$$\|\boldsymbol{x} - [\boldsymbol{x}]_t\| \leq rd/\vartheta_t, \quad \forall \boldsymbol{x} \in \mathbb{X}_t \tag{15}$$

where $[\boldsymbol{x}]_t$ denotes the closest point in $\mathbb{X}_t$ to $\boldsymbol{x}$. Now fix a constant $\delta > 0$ and an increasing sequence of positive constants $\{\pi_t\}_{t=1}^{\infty}$ satisfying $\sum_{t \geq 1} \pi_t^{-1} = 1$ (cf. Remark 2). Next, we introduce some Lemmas needed to bound the regret.

**Lemma 4 (Bogunovic et al. (2016, Proof of Theorem 4.2, cf. Appendix C.1))** *Pick* $\delta \in (0,1)$ *and set* $\beta_t = 2\ln\frac{\pi_t}{\delta}$, *where* $\sum_{t\geq 1}\pi_t^{-1} = 1$, $\pi_t > 0$. *Then we have*

$$\mathbb{P}\left\{|f_t(\boldsymbol{x}_t) - \tilde{\mu}_{\mathcal{D}_t}(\boldsymbol{x}_t)| \leq \sqrt{\beta_t}\tilde{\sigma}_{\mathcal{D}_t}(\boldsymbol{x}_t), \quad \forall t \geq 1\right\} \geq 1 - \delta. \tag{16}$$

**Lemma 5 (Bogunovic et al. (2016, Proof of Theorem 4.2, cf. Appendix C.1))** *Pick* $\delta \in (0,1)$ *and set* $\beta_t = 2\ln\frac{|\mathbb{X}_t|\pi_t}{\delta}$, *where* $\sum_{t\geq 1}\pi_t^{-1} = 1$, $\pi_t > 0$. *Then we have*

$$\mathbb{P}\left\{|f_t(\boldsymbol{x}_t) - \tilde{\mu}_{\mathcal{D}_t}(\boldsymbol{x}_t)| \leq \sqrt{\beta_t}\tilde{\sigma}_{\mathcal{D}_t}(\boldsymbol{x}_t), \quad \forall t \geq 1, \forall \boldsymbol{x} \in \mathbb{X}_t\right\} \geq 1 - \delta. \tag{17}$$

**Lemma 6 (Bogunovic et al. (2016, Proof of Theorem 4.2, cf. Appendix D))** *Pick* $\delta \in (0,1)$ *and set* $L_t = b_1\sqrt{\ln\frac{da_1\pi_t}{\delta}}$, *where* $\sum_{t\geq 1}\pi_t^{-1} = 1$, $\pi_t > 0$. *Then, with Assumption 3 we have*

$$\mathbb{P}\left\{\sup_{\boldsymbol{x}\in\mathbb{X}}\left|\frac{\partial f}{\partial x_j}\right| > L_t, \quad \forall t \geq 1, \forall \boldsymbol{x} \in \mathbb{X}_t, \forall j \in \{1,\ldots,d\}\right\} \leq 1 - \delta. \tag{18}$$

**Lemma 7** *Pick* $\delta \in (0,1)$ *and set* $\bar{y}_t = (b_0 + \sqrt{2}\sigma_{\mathrm{n}})\sqrt{\ln\frac{2(a_0+1)\pi_t}{\delta}}$, *where* $\sum_{t\geq 1}\pi_t^{-1} = 1$, $\pi_t > 0$. *Then we have*

$$\mathbb{P}\left\{|y_t| \leq \bar{y}_t, \forall t \geq 1\right\} \geq 1 - \delta. \tag{19}$$

**Proof:** We have that $|y_t| \leq |f_t(\boldsymbol{x}_t)| + |w_t|$ given Assumption 1. We can use Lemma 2 (with $t$ instead of $t_r$) with $\delta/2$ to bound $|w_t|$ for all time steps. Combined with Assumption 3 we have

$$\mathbb{P}\left\{|y_t| \leq L_f + \bar{w}_t\right\} \geq 1 - a_0 \exp\{-L_f^2/b_0^2\} - \delta/2. \tag{20}$$

Solving the remaining term with $\delta/2$ and using the union bound, we get $\bar{y}_t' = L_f + \bar{w}_t := b_0\sqrt{\ln\frac{2a_0\pi_t}{\delta}} + \bar{w}_t$. Here, we can further bound $b_0\sqrt{\ln\frac{2a_0\pi_t}{\delta}}$ with $b_0\sqrt{\ln\frac{2(a_0+1)\pi_t}{\delta}}$ using the monotonicity of the logarithm (recall that $a_0 > 0$). Since we have $\bar{w}_t = \sqrt{2}\sigma_{\mathrm{n}}\sqrt{\ln\frac{2\pi_t}{\delta}}$, we can use the same trick to state that $\bar{w}_t \leq \sqrt{2}\sigma_{\mathrm{n}}\sqrt{\ln\frac{2(a_0+1)\pi_t}{\delta}}$. To end, we can concatenate the terms obtaining $\bar{y}_t' \leq \bar{y}_t := (b_0 + \sqrt{2}\sigma_{\mathrm{n}})\sqrt{\ln\frac{2(a_0+1)\pi_t}{\delta}}$ proving the claim. $\square$

**Remark 3** *Note that Lemma 7 slightly differs from the bound used in Bogunovic et al. (2016). We explicitly consider the variance of the measurement noise to bound the observations in probability.*

For Theorem 1, we condition on four high probability events i.e., Lemma 4, Lemma 5, Lemma 6, and Lemma 7 each with with $\delta/4$. Conditioned on these events, the following Lemma holds.

**Lemma 8 (Bogunovic et al. (2016, Appendix D))** *Let Lemma 4, Lemma 5, and Lemma 6 each hold with probability* $1 - \delta/4$. *Pick* $\delta \in (0,1)$ *and set*

$$\beta_t = 2\ln\left(\frac{4\pi_t}{\delta}\right) + 2d\ln\left(rdt^2 b_1\sqrt{\ln(4a_1 d\pi_t/\delta)}\right),$$

*where* $\sum_{t\geq 1}\pi_t^{-1} = 1$, $\pi_t > 0$. *Let* $\vartheta_t = rdt^2 b_1\sqrt{\ln(2a_1 d\pi_t/\delta)}$ *and let* $[\boldsymbol{x}^*]_t$ *be the closest point in* $\mathbb{X}_t$ *to* $\boldsymbol{x}^*$. *Then we have*

$$\mathbb{P}\left\{|f_t(\boldsymbol{x}_t^*) - \tilde{\mu}_{\mathcal{D}_t}([\boldsymbol{x}_t^*]_t)| \leq \sqrt{\beta_t}\tilde{\sigma}_{\mathcal{D}_t}([\boldsymbol{x}_t^*]_t) + \frac{1}{t^2}, \forall t \geq 1\right\} \geq 1 - \delta. \tag{21}$$

In the following, we will explicitly highlight the usage of each Lemma. We first bound the instantaneous regret at a time step $t$ as

$$r_t = f_t(\boldsymbol{x}_t^*) - f_t(\boldsymbol{x}_t) \tag{22}$$

$$\leq \tilde{\mu}_{\mathcal{D}_t}([\boldsymbol{x}_t^*]_t) + \sqrt{\beta_t}\tilde{\sigma}_{\mathcal{D}_t}([\boldsymbol{x}_t^*]_t) + \frac{1}{t^2} - \tilde{\mu}_{\mathcal{D}_t}(\boldsymbol{x}_t) + \sqrt{\beta_t}\tilde{\sigma}_{\mathcal{D}_t}(\boldsymbol{x}_t). \quad \text{(Lemma 4, Lemma 8)} \tag{23}$$

This is identical to Bogunovic et al. (2016, Proof of Theorem 4.2) for R-GP-UCB. We now account for model mismatch between using the time-invariant model in the algorithm and the "true" time-varying posterior as

$$\tilde{\mu}_{\mathcal{D}_t}(\boldsymbol{x}) \leq \mu_{\mathcal{D}_t}(\boldsymbol{x}) + \Delta_{\mathcal{D}_t}^{(\mu)} \qquad \text{with} \quad \Delta_{\mathcal{D}_t}^{(\mu)} := \sup_{\boldsymbol{x} \in \mathbb{X}} \{|\tilde{\mu}_{\mathcal{D}_t}(\boldsymbol{x}) - \mu_{\mathcal{D}_t}(\boldsymbol{x})|\} \tag{24}$$

$$\tilde{\sigma}_{\mathcal{D}_t}(\boldsymbol{x}) \leq \sigma_{\mathcal{D}_t}(\boldsymbol{x}) + \Delta_{\mathcal{D}_t}^{(\sigma)} \qquad \text{with} \quad \Delta_{\mathcal{D}_t}^{(\sigma)} := \sup_{\boldsymbol{x} \in \mathbb{X}} \{|\tilde{\sigma}_{\mathcal{D}_t}(\boldsymbol{x}) - \sigma_{\mathcal{D}_t}(\boldsymbol{x})|\} . \tag{25}$$

We can thus further bound the regret as

$$r_t \leq \mu_{\mathcal{D}_t}([\boldsymbol{x}_t^*]_t) + \Delta_{\mathcal{D}_t}^{(\mu)}([\boldsymbol{x}_t^*]_t) + \sqrt{\beta_t} \left( \sigma_{\mathcal{D}_t}([\boldsymbol{x}_t^*]_t) + \Delta_{\mathcal{D}_t}^{(\sigma)}([\boldsymbol{x}_t^*]_t) \right) \tag{26}$$

$$- \mu_{\mathcal{D}_t}(\boldsymbol{x}_t) + \Delta_{\mathcal{D}_t}^{(\mu)}(\boldsymbol{x}_t) + \sqrt{\beta_t} \left( \sigma_{\mathcal{D}_t}(\boldsymbol{x}_t) + \Delta_{\mathcal{D}_t}^{(\sigma)}(\boldsymbol{x}_t) \right) + \frac{1}{t^2}. \tag{27}$$

Per definition of Algorithm 1, we have that through the optimization of its UCB-type acquisition function $\mu_{\mathcal{D}_t}([\boldsymbol{x}_t^*]_t) + \sqrt{\beta_t}\sigma_{\mathcal{D}_t}([\boldsymbol{x}_t^*]_t) \leq \mu_{\mathcal{D}_t}(\boldsymbol{x}_t) + \sqrt{\beta_t}\sigma_{\mathcal{D}_t}(\boldsymbol{x}_t)$, and, therefore,

$$r_t \leq 2\sqrt{\beta_t}\sigma_{\mathcal{D}_t}(\boldsymbol{x}_t) + \frac{1}{t^2} + \Delta_{\mathcal{D}_t}^{(\mu)}([\boldsymbol{x}_t^*]_t) + \Delta_{\mathcal{D}_t}^{(\mu)}(\boldsymbol{x}_t) + \sqrt{\beta_t} \left( \Delta_{\mathcal{D}_t}^{(\sigma)}([\boldsymbol{x}_t^*]_t) + \Delta_{\mathcal{D}_t}^{(\sigma)}(\boldsymbol{x}_t) \right). \tag{28}$$

What is left to bound is the model mismatch.[5] For this, we slightly adjust the bounds from Bogunovic et al. (2016, Lemma D.1.) to the setting where the block size can vary but is upper bounded by $\bar{N}$.

**Lemma 9 (Adjusted from Bogunovic et al. (2016, Lemma D.1.))** *Conditioned on the event in Lemma 7, we have for any block size $N \leq \bar{N}$ that the maximum model mismatch in the mean and variance between the time-varying GP model and the time-invariant GP model is a.s. bounded as*

$$\Delta_{\mathcal{D}_t}^{(\sigma)} \leq \sqrt{\left(3\sigma_{\mathrm{n}}^{-2} + \sigma_{\mathrm{n}}^{-4}\right) \bar{N}^3 \varepsilon} \tag{29}$$

$$\Delta_{\mathcal{D}_t}^{(\mu)} \leq \left(\sigma_{\mathrm{n}}^{-2} + \sigma_{\mathrm{n}}^{-4}\right) \bar{N}^3 \varepsilon \bar{y}_t. \tag{30}$$

**Proof:** For establishing the bounds on the model mismatch, we refer to the proof of Bogunovic et al. (2016, Lemma D.1.). If focuses on bounding (24) and (25) by explicitly bounding the difference in posterior distributions of the time-invariant updates in (4) and time-varying updates (5). The bounds in (29) and (30) are monotonic functions in the block length. Hence, the model mismatch is upper bounded by the maximum model mismatch with $N = \bar{N}$. □

With Lemma 9, we can summarize the bound on the model mismatch at each time step as

$$2\Delta_{\mathcal{D}_t}^{(\mu)} + 2\sqrt{\beta_t}\Delta_{\mathcal{D}_t}^{(\sigma)} \leq 2\left(\sigma_{\mathrm{n}}^{-2} + \sigma_{\mathrm{n}}^{-4}\right) \bar{N}^3 \varepsilon \bar{y}_t + 2\sqrt{\beta_t \left(3\sigma_{\mathrm{n}}^{-2} + \sigma_{\mathrm{n}}^{-4}\right) \bar{N}^3 \varepsilon} =: \phi_t(\varepsilon, \bar{N}). \tag{31}$$

For completeness, we follow the steps in Srinivas et al. (2010, Proof of Theorem 2) and Bogunovic et al. (2016, Proof of Theorem 4.2 (R-GP-UCB)) to bound the regret following Definition 1 as

$$R_T := \sum_{t=1}^{T} r_t \leq \sum_{t=1}^{T} 2\sqrt{\beta_t}\sigma_{\mathcal{D}_t}(\boldsymbol{x}_t) + \frac{1}{t^2} + \phi_t(\varepsilon, \bar{N}). \tag{32}$$

With $\sum_{t=1}^{\infty} 1/t^2 = \pi^2/6 \leq 2$ and using Cauchy-Schwarz with $\left(\sum_{t=1}^{T} \boldsymbol{x}_t\right)^2 \leq T \sum_{t=1}^{T} \boldsymbol{x}_t^2$ we obtain

$$R_T \leq \sqrt{T \sum_{t=1}^{T} 4\beta_t \sigma_{\mathcal{D}_t}^2(\boldsymbol{x}_t) + 2} + \sum_{t=1}^{T} \left( 2\left(\sigma_{\mathrm{n}}^{-2} + \sigma_{\mathrm{n}}^{-4}\right) \bar{N}^3 \varepsilon \bar{y}_t + 2\sqrt{\beta_t \left(3\sigma_{\mathrm{n}}^{-2} + \sigma_{\mathrm{n}}^{-4}\right) \bar{N}^3 \varepsilon} \right). \tag{33}$$

---

[5]We noticed a minor mistake in Bogunovic et al. (2016, Theorem 4.2). There, it should be $2 \cdot T\psi_T(N, \epsilon)$ in Eq. (17) as model differences in mean and variance have to be bounded twice as it is the case for the variance in our proof (cf. Bogunovic et al. (2016, Proof of Theorem 4.2), Eq. (72)). There are no consequences from the missing constant but it might be helpful for future work to state it here explicitly.

Furthermore, using the same arguments as in Bogunovic et al. (2016, Proof of Theorem 4.2 (R-GP-UCB)) and Srinivas et al. (2010, Proof of Lemma 5.3 (GP-UCB)) on the information gain within a block of size $M$ is $I(y_M; f_M) = \frac{1}{2} \sum_{t=1}^{M} \ln\left(1 + \sigma_n^{-2} \sigma_{\mathcal{D}_t}^2(\boldsymbol{x}_t)\right) \leq \gamma_M$ with $\gamma_M = \max_{\boldsymbol{x}_1, \ldots \boldsymbol{x}_M} I(y_M; f_M)$ yields

$$\sum_{t=1}^{T} 4\beta_t \sigma_{\mathcal{D}_t}^2(\boldsymbol{x}_t) \leq 4\beta_T \sigma_n^2 \sum_{t=1}^{T} \left(\sigma_n^{-2} \sigma_{\mathcal{D}_t}^2(\boldsymbol{x}_t)\right) \tag{34}$$

$$\leq 4\beta_T \sigma_n^2 C_2 \sum_{t=1}^{T} \ln\left(1 + \sigma_n^{-2} \sigma_{\mathcal{D}_t}^2(\boldsymbol{x}_t)\right) \tag{35}$$

with $C_2 = \sigma_n^{-2} / \ln(1 + \sigma_n^{-2}) \geq 1$, since $s^2 \leq C_2 \ln(1 + s^2)$ for $s \in [0, \sigma_n^2]$, and $\sigma_n^{-2} \sigma_{\mathcal{D}_t}^2(\boldsymbol{x}_t) \leq \sigma_n^{-2} k(\boldsymbol{x}_t, \boldsymbol{x}_t) \leq \sigma_n^{-2}$ due to the bounded kernel function in Assumption 2 (also see Srinivas et al. (2010, Proof of Lemma 5.4), we get

$$\sum_{t=1}^{T} 4\beta_t \sigma_{\mathcal{D}_t}^2(\boldsymbol{x}_t) \leq 8\beta_T \sigma_n^2 C_2 \frac{T}{\underline{N}} I(y_{\bar{N}}; f_{\bar{N}}) \leq 8\beta_T \sigma_n^2 C_2 \frac{T}{\underline{N}} \gamma_{\bar{N}}. \tag{36}$$

Here, the lower reset bound $\underline{N}$ becomes relevant with the following argument. With block lengths as $N \in [\underline{N}, \bar{N}]$, the maximum number of blocks one could obtain until $T$ is bounded by $\frac{T}{\underline{N}}$ (assuming this is an integer for now). In each of the blocks, the maximum information gain is bounded by $\gamma_{\bar{N}}$ since the information gain is non decreasing. We assumed that $\frac{T}{\underline{N}}$ is an integer as the number of block in $T$ and we can upper this with $\left(\frac{T}{\underline{N}} + 1\right)$. We therefore have

$$R_T \leq \sqrt{C_1 T \beta_T \left(\frac{T}{\underline{N}} + 1\right) \gamma_{\bar{N}}} + 2 + \sum_{t=1}^{T} 2\left(\sigma_n^{-2} + \sigma_n^{-4}\right) \bar{N}^3 \varepsilon \bar{y}_t + 2\sqrt{\beta_t \left(3\sigma_n^{-2} + \sigma_n^{-4}\right) \bar{N}^3 \varepsilon} \tag{37}$$

with $C_1 = 8 / \ln(1 + \sigma_n^{-2})$. Further bounding the sum over maximum model mismatch yields

$$R_T \leq \sqrt{C_1 T \beta_T \left(\frac{T}{\underline{N}} + 1\right) \gamma_{\bar{N}}} + 2 + T \underbrace{\left(2\left(\sigma_n^{-2} + \sigma_n^{-4}\right) \bar{N}^3 \varepsilon \bar{y}_T + 2\sqrt{\beta_T \left(3\sigma_n^{-2} + \sigma_n^{-4}\right) \bar{N}^3 \varepsilon}\right)}_{\phi_T(\varepsilon, \bar{N}) \text{ in Theorem 1}}.$$

$\square$

## F.2 Proof of Corollary 1

**Restated Corollary 1** *Let the assumptions in Theorem 1 hold. Given a lower and upper bound on the true $\varepsilon$ such that $\varepsilon \in [\underline{\varepsilon}, \bar{\varepsilon}] \subset (0, 1)$, there exist $\lambda_1 \in (0, 1]$ and $\lambda_2 \in [1, \frac{1}{\varepsilon})$ such that $\underline{\varepsilon} = \lambda_1 \cdot \varepsilon$ and $\bar{\varepsilon} = \lambda_2 \cdot \varepsilon$. For the SE kernel, set $\underline{N} = \Theta(\min\{T, \bar{\varepsilon}^{-1/4}\})$ and $\bar{N} = \Theta(\min\{T, \underline{\varepsilon}^{-1/4}\})$. Then,*

$$R_T = \tilde{\mathcal{O}}\left(\max\left\{\sqrt{T}, \lambda_2^{1/8} \varepsilon^{1/8} T, \lambda_1^{-3/8} \varepsilon^{1/8} T, \lambda_1^{-3/4} \varepsilon^{1/4} T\right\}\right). \tag{38}$$

*For the Matèrn kernel ($\nu > 2$), define $\xi = \frac{d(d+1)}{2\nu + d(d+1)}$ and set $\underline{N} = \Theta(\min\{T, \bar{\varepsilon}^{-1/(4-\xi)}\})$ and $\bar{N} = \Theta(\min\{T, \underline{\varepsilon}^{-1/(4-\xi)}\})$. Then,*

$$R_T = \tilde{\mathcal{O}}\left(\max\left\{\sqrt{T^{1-\xi}}, \lambda_1^{-\frac{\xi}{2(4-\xi)}} \lambda_2^{\frac{1}{2(4-\xi)}} \varepsilon^{\frac{1-\xi}{2(4-\xi)}} T, \lambda_1^{-\frac{3}{2(4-\xi)}} \varepsilon^{\frac{1-\xi}{2(4-\xi)}} T, \lambda_1^{-\frac{3}{4-\xi}} \varepsilon^{\frac{1-\xi}{4-\xi}} T\right\}\right). \tag{39}$$

**Proof:** First recall the bound from Theorem 1 and divide it into three terms to increase readability.

$$R_T \leq \underbrace{\sqrt{C_1 T \beta_T \left(\frac{T}{\underline{N}} + 1\right) \gamma_{\bar{N}}}}_{\text{Term 1}} + 2 + \underbrace{2T\sqrt{\beta_T \left(3\sigma_n^{-2} + \sigma_n^{-4}\right) \bar{N}^3 \varepsilon}}_{\text{Term 2}} + \underbrace{2T\left(\sigma_n^{-2} + \sigma_n^{-4}\right) \bar{N}^3 \varepsilon \bar{y}_T}_{\text{Term 3}}$$

In the following, we will first consider the SE kernel and afterwards the general Matérn kernel. We will bound each term separately for both kernels. The proof builds on Bogunovic et al. (2016, Appendix E) but extends it to the setting where only lower and upper bounds on $\varepsilon$ are known. In the following, let $\mathcal{I}_T(z)$ be the closest integer in $\{1, \ldots, T\}$ which is closest to $z \in \mathbb{R}$.

**Squared Exponential Kernel**

**Term 1** For the SE kernel, Srinivas et al. (2010, Theorem 5) shows that $\gamma_{\bar{N}} = \mathcal{O}(\ln \bar{N}) = \tilde{\mathcal{O}}(1)$. Setting $\underline{N} = \mathcal{I}_T(\bar{\varepsilon}^{-1/4})$, we obtain upon substitution

$$\sqrt{C_1 T \beta_T \left(\frac{T}{\underline{N}} + 1\right) \gamma_{\bar{N}}} = \tilde{\mathcal{O}}\left(\sqrt{T \frac{T}{\bar{\varepsilon}^{-1/4}}}\right) = \tilde{\mathcal{O}}\left(T \sqrt{\frac{1}{(\lambda_2 \varepsilon)^{-1/4}}}\right) = \tilde{\mathcal{O}}\left(\lambda_2^{1/8} \varepsilon^{1/8} T\right). \tag{40}$$

**Term 2** For the second term, we have upon substitution of $\bar{N} = \mathcal{I}_T(\varepsilon^{-1/4})$

$$2T \sqrt{\beta_T \left(3\sigma_{\mathrm{n}}^{-2} + \sigma_{\mathrm{n}}^{-4}\right) \bar{N}^3 \varepsilon} = \tilde{\mathcal{O}}\left(T \sqrt{\varepsilon^{-3/4} \varepsilon}\right) = \tilde{\mathcal{O}}\left(T \sqrt{(\lambda_1 \varepsilon)^{-3/4} \varepsilon}\right) = \tilde{\mathcal{O}}\left(\lambda_1^{-3/8} \varepsilon^{1/8} T\right). \tag{41}$$

**Term 3** For the third term, we have upon substitution of $\bar{N} = \mathcal{I}_T(\varepsilon^{-1/4})$

$$2T \left(\sigma_{\mathrm{n}}^{-2} + \sigma_{\mathrm{n}}^{-4}\right) \bar{N}^3 \varepsilon \bar{y}_T = \tilde{\mathcal{O}}\left(T \varepsilon^{-3/4} \varepsilon\right) = \tilde{\mathcal{O}}\left(T (\lambda_1 \varepsilon)^{-3/4} \varepsilon\right) = \tilde{\mathcal{O}}\left(\lambda_1^{-3/4} \varepsilon^{1/4} T\right). \tag{42}$$

From these results, we can further see that for the scaling to be sub-linear $\varepsilon$ has to be sufficiently small relative to the time horizon $T$ and the lower and upper bounds on $\varepsilon$ have to be sufficiently close. Specifically, if $\varepsilon < \min\left\{\frac{1}{\lambda_2 T^4}, \frac{\lambda_1^3}{T^4}\right\}$ we obtain sub-linear scaling, i.e., $\tilde{\mathcal{O}}(\sqrt{T})$.

**Matérn Kernel ($\nu > 2$)**

**Term 1** For the Matérn Kernel kernel, Srinivas et al. (2010, Theorem 5) shows that $\gamma_{\bar{N}} = \mathcal{O}(\bar{N}^\xi \ln \bar{N}) = \tilde{\mathcal{O}}(\bar{N}^\xi)$ where $\xi = \frac{d(d+1)}{2\nu + d(d+1)}$. Setting $\underline{N} = \mathcal{I}_T(\bar{\varepsilon}^{-1/(4-\xi)})$ and $\bar{N} = \mathcal{I}_T(\varepsilon^{-1/(4-\xi)})$, we have upon substitution

$$\sqrt{C_1 T \beta_T \left(\frac{T}{\underline{N}} + 1\right) \gamma_{\bar{N}}} = \tilde{\mathcal{O}}\left(\sqrt{T \frac{T}{\bar{\varepsilon}^{-1/(4-\xi)}} \varepsilon^{-\xi/(4-\xi)}}\right) \tag{43}$$

$$= \tilde{\mathcal{O}}\left(T \sqrt{\frac{1}{(\lambda_2 \varepsilon)^{-1/(4-\xi)}} (\lambda_1 \varepsilon)^{-\xi/(4-\xi)}}\right) \tag{44}$$

$$= \tilde{\mathcal{O}}\left(\lambda_1^{-\frac{\xi}{2(4-\xi)}} \lambda_2^{\frac{1}{2(4-\xi)}} \varepsilon^{\frac{1-\xi}{2(4-\xi)}} T\right). \tag{45}$$

**Term 2** For the second term, we have upon substitution of $\bar{N} = \mathcal{I}_T(\varepsilon^{-1/(4-\xi)})$

$$2T \sqrt{\beta_T \left(3\sigma_{\mathrm{n}}^{-2} + \sigma_{\mathrm{n}}^{-4}\right) \bar{N}^3 \varepsilon} = \tilde{\mathcal{O}}\left(T \sqrt{\varepsilon^{-3/(4-\xi)} \varepsilon}\right) \tag{46}$$

$$= \tilde{\mathcal{O}}\left(T \sqrt{(\lambda_1 \varepsilon)^{-3/(4-\xi)} \varepsilon}\right) \tag{47}$$

$$= \tilde{\mathcal{O}}\left(\lambda_1^{-\frac{3}{2(4-\xi)}} \varepsilon^{\frac{1-\xi}{2(4-\xi)}} T\right). \tag{48}$$

**Term 3** For the third term, we have upon substitution of $\bar{N} = \mathcal{I}_T(\varepsilon^{-1/(4-\xi)})$

$$2T \left(\sigma_{\mathrm{n}}^{-2} + \sigma_{\mathrm{n}}^{-4}\right) \bar{N}^3 \varepsilon \bar{y}_T = \tilde{\mathcal{O}}\left(T \varepsilon^{-3/(4-\xi)} \varepsilon\right) \tag{49}$$

$$= \tilde{\mathcal{O}}\left(T (\lambda_1 \varepsilon)^{-3/(4-\xi)} \varepsilon\right) \tag{50}$$

$$= \tilde{\mathcal{O}}\left(\lambda_1^{-\frac{3}{4-\xi}} \varepsilon^{\frac{1-\xi}{4-\xi}} T\right). \tag{51}$$

Similar to the SE kernel, we can also identify the sub-linear region for the Matérn kernel. Specifically, if $\varepsilon < \min\left\{\frac{\lambda_1^\xi}{\lambda_2^{1-\xi}T^{4-\xi}}, \frac{\lambda_1^{3(1-\xi)}}{T^{4-\xi}}\right\}$ we obtain sub-linear scaling, i.e., $\tilde{\mathcal{O}}(\sqrt{T^{1-\xi}})$. We can furthermore nicely observe that for $\nu \to \infty$ (therefore $\xi \to 0$), the bounds for Terms 1-3 as well as the sub-linear region converges to the bounds of the SE kernel. $\qquad\square$

## G  Proof of Lemma 2 and Lemma 3

**Restated Lemma 2**  *Pick $\delta \in (0, 1)$ and set $\bar{w}_{t_r}^2 = 2\sigma_n^2 \ln \frac{\pi_{t_r}}{\delta}$, where $\sum_{t_r \geq 1} \pi_{t_r}^{-1} = 1$, $\pi_{t_r} > 0$. Then, the noise sequence in Assumption 1 obtained since the last reset satisfies $|w_{t_r}| \leq \bar{w}_{t_r}$ for all $t_r \geq 1$ with probability at least $1 - \delta$.*

**Proof:**  Since $w_{t_r} \sim \mathcal{N}(0, \sigma_n^2)$, we can use a standard bound on the tail probability of the normal distribution $\mathbb{P}\{|r| > s\} \leq \exp\left\{-s^2/(2\sigma^2)\right\}$ with $r \sim \mathcal{N}(0, \sigma^2)$ to obtain $\mathbb{P}\{|w_{t_r}| \leq \bar{w}_{t_r}\} \geq 1 - \exp\left\{-\bar{w}_{t_r}^2/(2\sigma_n^2)\right\}$. Solving for $\bar{w}_t$ with $\delta/\pi_{t_r}$ using the same arguments as in the proof of Srinivas et al. (2010, Lemma 5.5) and taking the union bound over all time steps yields the results. $\qquad\square$

**Restated Lemma 3**  *Let $f_t(\boldsymbol{x})$ follow Assumption 2 with $\varepsilon = 0$. Pick $\delta_B \in (0, 1)$ and set $\rho_{t_r} = 2\ln \frac{2\pi_{t_r}}{\delta_B}$, where $\sum_{t_r \geq 1} \pi_{t_r}^{-1} = 1$, $\pi_{t_r} > 0$. Also set $\bar{w}_{t_r}^2 = 2\sigma_n^2 \ln \frac{2\pi_{t_r}}{\delta_B}$. Then, observations $y_t$ under Assumption 1 satisfy $|y_t - \mu_{\mathcal{D}_t}(\bar{\boldsymbol{x}}_t)| \leq \sqrt{\rho_{t_r}}\sigma_{\mathcal{D}_t}(\boldsymbol{x}_t) + \bar{w}_{t_r}$ for all $t_r \geq 1$ with probability at least $1 - \delta_B$.*

**Proof:**  From Lemma 1 with $\delta_B/2$ and incorporating the noise, we have

$$|f_t(\boldsymbol{x}_t) + w_{t_r} - \mu_{\mathcal{D}_t}(\boldsymbol{x}_t)| \leq \sqrt{\rho_{t_r}}\sigma_{\mathcal{D}_t}(\boldsymbol{x}_t) + |w_{t_r}|$$
$$|y_t - \mu_{\mathcal{D}_t}(\boldsymbol{x}_t)| \leq \sqrt{\rho_{t_r}}\sigma_{\mathcal{D}_t}(\boldsymbol{x}_t) + |w_{t_r}|. \tag{52}$$

with $\rho_{t_r} = 2\ln \frac{2\pi_{t_r}}{\delta_B}$. Using Lemma 2 with $\delta_B/2$ and taking the union bound yields the result. $\qquad\square$

## H  Hyperparameters

To ensure the reproducibility of our results, we give a list of all hyperparameters used in any simulation performed for this paper. All experiments in Section 6.2 were conducted on a 2021 MacBook Pro with an Apple M1 Pro chip and 16GB RAM. The Monte Carlo simulations in Section 5 were conducted over 3 days on a compute cluster. The temperature data set is publicly available at `http://db.csail.mit.edu/labdata/labdata.html`. The code will be published upon acceptance and is also part of the supplementary material of the submission.

Table 5: Hyperparameters of the Monte Carlo Simulation for ET-GP-UCB (cf. Figure 2).

| Hyperparameters | Monte Carlo Simulation |
|---|---|
| dimension $d$ | 2 |
| compact set $\mathbb{X}$ | $[0, 1]^2$ |
| lengthscales $l$ | 0.2 |
| time horizon $T$ | 200 |
| noise variance $\sigma_n^2$ | 0.02 |
| number of i.i.d. runs | 30 000 |
| event trigger parameter in Lemma 3 $\delta_B$ | 0.1 |
| rate of change $\varepsilon$ | $\{0.03, 0.1\}$ |

Table 6: Hyperparameters for the within-model comparisons (cf. Figure 7).

| Algorithm | Hyperparameters | Correct rate of change | Miss. rate of change |
|---|---|---|---|
| all | dimension $d$ | 2 | 2 |
| | compact set $\mathbb{X}$ | $[0,1]^2$ | $[0,1]^2$ |
| | lengthscales $l$ | 0.2 | 0.2 |
| | time horizon $T$ | 400 | 400 |
| | noise variance $\sigma_{\mathrm{n}}^2$ | 0.02 | 0.02 |
| | $\beta_t$ approximation parameters | $c_1 = 0.4, c_2 = 4$ | $c_1 = 0.4, c_2 = 4$ |
| | number of i.i.d. runs | 50 | 50 |
| UI-TVBO | rate of change $\hat{\sigma}_{\mathrm{w}}^2$ | $\{0.01, 0.03, 0.05\}$ | $\{0.001, 0.2\}$ |
| TV-GP-UCB | rate of change $\varepsilon$ | $\{0.01, 0.03, 0.05\}$ | $\{0.001, 0.2\}$ |
| R-GP-UCB | reset time $N_{\mathrm{const}}$ | $\{38, 29, 26\}$ | $\{68, 17\}$ |
| ET-GP-UCB | event trigger parameter $\delta_{\mathrm{B}}$ | 0.1 | 0.1 |

Table 7: Hyperparameters for the temperature dataset (cf. Figure 3).

| Algorithm | Hyperparameters | Temperature Dataset |
|---|---|---|
| all | dimension $d$ | 1 |
| | compact set $\mathbb{X}$ | 46 arms |
| | kernel $k$ | empirical kernel |
| | time horizon $T$ | 286 |
| | noise variance $\sigma_{\mathrm{n}}^2$ | 0.01 |
| | $\beta_t$ approximation parameters | $c_1 = 0.8, c_2 = 0.4$ |
| | number of i.i.d. runs | 50 |
| UI-TVBO | rate of change $\hat{\sigma}_{\mathrm{w}}^2$ | 0.03 |
| TV-GP-UCB | rate of change $\varepsilon$ (see Bogunovic et al. (2016)) | 0.03 |
| R-GP-UCB | reset time $N_{\mathrm{const}}$ (see Bogunovic et al. (2016)) | 15 |
| ET-GP-UCB | event trigger parameter in Lemma 3 $\delta_{\mathrm{B}}$ | 0.1 |

Table 8: Hyperparameters for the policy search experiment (cf. Figure 3 (c)).

| Algorithm | Hyperparameters | Policy search |
|---|---|---|
| all | dimension $d$ | 4 |
| | compact set $\mathbb{X}$ | lower bound to $[-3, -6, -50, -4]$ upper bound to $[-2, -4, -25, -2]$ scaling factors $[1/8, 1/4, 3, 1/8]$ |
| | lengthscales $l$ | gamma prior $\mathcal{G}(15, 10/3)$ |
| | time horizon $T$ | 300 |
| | noise variance $\sigma_{\mathrm{n}}^2$ | 0.02 |
| | $\beta_t$ approximation parameters | $c_1 = 0.8, c_2 = 4$ |
| | number of i.i.d. runs | 50 |
| UI-TVBO | rate of change $\hat{\sigma}_{\mathrm{w}}^2$ | 0.03 |
| TV-GP-UCB | rate of change $\varepsilon$ | 0.03 |
| R-GP-UCB | reset time $N_{\mathrm{const}}$ induced by rate of change $\varepsilon$ | 29 |
| UI-TVBO | rate of change $\hat{\sigma}_{\mathrm{w}}^2$ | 0.03 |
| ET-GP-UCB | event trigger parameter in Lemma 3 $\delta_{\mathrm{B}}$ | 0.1 |

Table 9: Ablation on Impact of Reset Frequency vs. Timing (cf. Figure 4).

| Algorithm | Hyperparameters | Sensitivity Analysis |
|---|---|---|
| all | dimension $d$ | 2 |
| | compact set $\mathbb{X}$ | $[0,1]^2$ |
| | kernel $k$ | fixed, $l = 0.2$ |
| | time horizon $T$ | 400 |
| | noise variance $\sigma_n^2$ | 0.02 |
| | $\beta_t$ approximation parameters | $c_1 = 0.4, c_2 = 4$ |
| | number of i.i.d. runs | 50 |
| R-GP-UCB | reset time $N_{\text{const}}$ | $\{38, 29, 26\}$ |
| R-GP-UCB* | reset time $N_{\text{const}}$ as average from ET-GP-UCB | $\{100, 45, 32\}$ |
| ET-GP-UCB | event trigger parameter in Lemma 3 $\delta_B$ | $\{0.1, 0.1, 0.1\}$ |

Table 10: Hyperparameters for the synthetic experiments in Deng et al. (2022).

| Algorithm | Hyperparameters | Synthetic Experiments |
|---|---|---|
| | dimension $d$ | 1 |
| | compact set $\mathbb{X}$ | $[0,1]^2$ |
| | kernel $k$ | fixed, $l = 0.2$ |
| | time horizon $T$ | 500 |
| | noise variance $\sigma_n^2$ | $\{0.01, 1\}$ |
| | number of i.i.d. runs | 20 |
| ET-GP-UCB | event trigger parameter in Lemma 3 $\delta_B$ | 0.1 |

Table 11: Hyperparameters for the stock market data experiment in Deng et al. (2022).

| Algorithm | Hyperparameters | Stock Market Data |
|---|---|---|
| | dimension $d$ | 1 |
| | compact set $\mathbb{X}$ | 48 arms |
| | kernel $k$ | empirical kernel |
| | time horizon $T$ | 823 |
| | noise variance $\sigma_n^2$ | $\{0.01, 300\}$ |
| | number of i.i.d. runs | 20 |
| ET-GP-UCB | event trigger parameter in Lemma 3 $\delta_B$ | 0.1 |

Table 12: Hyperparameters for the sensitivity analysis (cf. Table 4).

| Algorithm | Hyperparameters | Sensitivity Analysis |
|---|---|---|
| all | dimension $d$ | 2 |
| | compact set $\mathbb{X}$ | $[0,1]^2$ |
| | kernel $k$ | fixed, $l = 0.2$ |
| | time horizon $T$ | 400 |
| | noise variance $\sigma_n^2$ | 0.02 |
| | $\beta_t$ approximation parameters | $c_1 = 0.4, c_2 = 4$ |
| | number of i.i.d. runs | 50 |
| ET-GP-UCB | event trigger parameter in Lemma 3 $\delta_B$ | $\{0.005, 0.01, 0.05, 0.1, 0.5\}$ |

