# OpenReview forum: "Event-Triggered Time-Varying Bayesian Optimization"
_TMLR — Accepted by TMLR_

### Review · Reviewer_TDqt · 2024-11-05

**Summary Of Contributions:**

This paper proposes a novel event-triggered algorithm (ET-GP-UCB) to address the time-varying Bayesian optimization (TVBO) problem. Unlike existing approaches, ET-GP-UCB treats the optimization as static until a change in the objective function is detected, at which point it resets the dataset. This enables the algorithm to adapt dynamically without requiring prior knowledge of the objective’s rate of change. The authors provide theoretical analysis for the method and demonstrate its superior performance on synthetic and real-world datasets.

**Audience:**

Yes

**Broader Impact Concerns:**

No.

**Claims And Evidence:**

No

**Requested Changes:**

* The proposed event trigger strategy should be compared with performance-based reset strategies.
* More benchmarks should be used for evaluation, e.g., synthetic functions and real-world applications with more dimensions or even a changing change rate ($\epsilon$ is not a constant).
* More details and more experiments about how event trigger hyperparameters affect the optimization process should be provided.

**Strengths And Weaknesses:**

## Strengths
* The event-triggered idea is easy to understand.
* The paper presents a theoretical analysis with clear regret bounds, providing a foundation for the algorithm’s effectiveness.
* Many discussions including ablation study are provided.

## Weaknesses
* Although theoretical analysis is provided, it relies heavily on the posterior variance of a Gaussian process (GP) model, assuming exact adherence to GP priors. In practice, it is unlikely that data would align perfectly with such priors. Consequently, the event-trigger mechanism could signal unnecessary resets in static optimization scenarios, raising concerns about the algorithm’s robustness in real-world applications.
*  Based on the aforementioned concern in the real-world scenarios, the hyperparameter settings of event trigger is important and should influence the performance. In Appendix B, the authors show how the hyperparameter $\delta_B$ affect the regret performance. However, it does not provide enough details about the experiments, e.g. the benchmarks and the number of iterations. It is also important to provide line charts showing how the regret changes over time instead of only the final regret, as changing $\delta_B$ might affect the convergence speed but has limited impact on final regrets.
* The baselines selected for comparison are somewhat limited in complexity. A more robust comparison could include a simple and adaptive reset strategies based on actual optimization performance, such as the reset strategy in TuRBO [1], where resets are triggered when consecutive iterations fail to yield improvement.
* The proposed method is validated only on synthetic and specific real data, lacking experimental support for a broader range of domains or more complex scenarios, such as those with dimensions greater than 5.

[1] Scalable Global Optimization via Local Bayesian Optimization, Neurips 2019.

---

> ### Author Response · Authors · 2024-11-15
>
> Thank you for taking the time to review our paper. We greatly appreciate it and hope to address all questions and concerns in the following.
>
> ## Addressing the Requested Changes
>
> > 1. The proposed event trigger strategy should be compared with performance-based reset strategies.
>
> Thank you for the suggestion. Performance-based reset strategies are definitely an interesting research direction for the time-varying setting and can be well formulated within the framework we propose in Definition 2.
> While performance-based reset algorithms seem straightforward, applying them to a dynamic setting would require some non-trivial modifications.
>
> More precisely, resetting once performance no longer improves could result in a significant number of resets over time since exploration can, of course, result in non-improving queries.
> Therefore, as you state, one would have to introduce more hyperparameters that specify after how many "failures" to improve (similar to TuRBO) the algorithm should reset. However, being able to improve is not necessarily a reliable trigger for changes since a function could change to be higher at the query point. A performance-based trigger would only detect changes in one direction.
> Our current event trigger can also be considered a performance-based trigger as we only consider the output, but through (11), we check for changes in both directions, i.e., unexpected high or unexpected low performance.
> Lastly, unlike TuRBO, a reset significantly impacts performance since all data is discarded (while shrinking the trust region for TuRBO is less drastic).
> We believe that while performance-based resets hold promise, they fall beyond the scope of this work and would require further in-depth investigation and algorithm design for the dynamic settings to yield well-performing approaches.
>
> > 2. More benchmarks should be used for evaluation, e.g., synthetic functions and real-world applications with more dimensions or even a changing change rate (epsilon is not a constant).
>
> We appreciate the suggestion for more extensive benchmarking. Given the implicit page limit of 12 pages, we believe that our current empirical evaluation is already quite exhaustive. In Section 6.5 (previous version, Section 6.3 in the revised version), we show results up to eight input dimensions, which is significantly higher than in previous work (e.g., Bogunovic et al. 2016). We also include two benchmark problems in the main paper and three additional problems in Appendix A.3.
>
> That said, adding a benchmark with a changing rate of change is very interesting.
> This can further highlight the adaptive nature of our event-triggered approach.
> We have included an additional experiment in the revised version (Appendix A.2) of our paper in which we change the rate of change after some time step.
> We initialize all algorithms that require epsilon as an input to their algorithm with the rate of change at time step 0.
> In the results in Figure 6, the adaptive nature of our algorithm becomes evident.
> After the change in the rate of change $\varepsilon$, ET-GP-UCB starts to increase the reset frequency adjusting to the higher rate of change in the unknown objective.
> In contrast, TV-GP-UCB slowly starts to diverge over time as the model no longer uses the true $\varepsilon$.
> While the difference for 500 time steps is not as big, there is a clear diverging trend.
>
> > 3. More details and more experiments about how event trigger hyperparameters affect the optimization process should be provided.
>
> Thank you for this suggestion. In the revised version, we include the cumulative regret over time in Figure 11 to accompany the results in Table 4 and highlight the robustness of $\delta_B$ with respect to the optimization behavior. We can observe that the convergence behavior is not changed. For all deltas, we used 50 independent seeds.

---

> > ### Comment · Reviewer_TDqt · 2024-11-16
> > **Reply to Authors**
> >
> > Thanks to the authors for their response. However, the comment only partly addressed my concern.
> > * **Robustness to Priors.** My main concern is that the proposed algorithm relies on prior GP assumptions (the event-trigger mechanism might signal unnecessary resets in static optimization scenarios). Many past case studies have shown discrepancies between theoretical priors and actual optimization scenarios, even the original GP-UCB requires proper hyperparameter settings.  I hoped for more empirical evidence to demonstrate the algorithm’s robustness rather than avoiding this issue. For example, the authors could observe whether the proposed event-trigger mechanism will signal unnecessary resets in static optimization scenarios.
> > * **Baseline Comparison.** The reset strategy based on performance observations, such as the one used in TurBO, is a natural approach in practical settings. Implementing such a strategy by setting suitable hyper-parameters should not be difficult. I do not understand why the authors chose not to compare their method against this type of baseline. Such a comparison would also help validate the proposed algorithm's superiority and alleviate my primary (aforementioned) concern.
> > * **Details of Figure 11.** Figure 11 still lacks experimental details, such as which benchmarks/datasets are used in Figure 11. Additionally, could the authors provide an intuitive explanation of why hyperparameter settings have such a minimal impact on performance?

---

> > > ### Author Response · Authors · 2024-11-21
> > >
> > > Thank you for engaging in the discussion, we really appreciate it.
> > >
> > > > Robustness to Priors. My main concern is that the proposed algorithm relies on prior GP assumptions (the event-trigger mechanism might signal unnecessary resets in static optimization scenarios). Many past case studies have shown discrepancies between theoretical priors and actual optimization scenarios, even the original GP-UCB requires proper hyperparameter settings. I hoped for more empirical evidence to demonstrate the algorithm’s robustness rather than avoiding this issue. For example, the authors could observe whether the proposed event-trigger mechanism will signal unnecessary resets in static optimization scenarios.
> > >
> > > We understand this concern and are, therefore, now included in experiments on standard benchmark functions.
> > > Specifically, we chose the two-dimensional Ackley function as well as the six-dimensional Hartmann function.
> > > To include the time variation, we further induced a shift in the optimum after some time steps.
> > > We chose the timing of this shift to be after a significant amount of time steps to also be able to evaluate the static behaviour.
> > > The results are in Appendix A.4.
> > > We can observe that both ET-GP-UCB variants had some resets at the beginning of the optimization, which we attribute to the mismatch between the prior and the objective you mentioned.
> > > However, we demonstrate that the influence on the final performance (before the shift) is not significant for both benchmarks.
> > > As expected, the time-varying baselines can cope with the shift in the optimum whereas standard GP-UCB can not.
> > > The additional experiments, however, yielded something even more interesting.
> > > On Hartmann in Figure 10, we can observe that the ET-GP-UCB variant with backtracking significantly outperformed the static GP-UCB even before the change!
> > > We believe that by discarding some of the past data, ET-GP-UCB with backtracking can learn better lengthscales relevant for finding the optimum.
> > > The bad performance of GP-UCB is also tightly linked to the mismatch between the prior and the objective.
> > > This additional result is intriguing because it suggests that the adaptive approaches developed for tackling time-varying optimization problems also show promise for static settings.
> > > We are happy to move this result to the main body as we believe it can be of significant interest to the BO community. Given that we have already surpassed the 12-page limit of a standard paper, however, we would like the editor's opinion on this.
> > >
> > >
> > > > Baseline Comparison. The reset strategy based on performance observations, such as the one used in TurBO, is a natural approach in practical settings. Implementing such a strategy by setting suitable hyper-parameters should not be difficult. I do not understand why the authors chose not to compare their method against this type of baseline. Such a comparison would also help validate the proposed algorithm's superiority and alleviate my primary (aforementioned) concern.
> > >
> > > We have implemented a TuRBO-like event trigger to reset once the performance no longer improves for some time steps $\tau_{fail}$.
> > > Here, we use the same heuristic proposed in the TuRBO paper [R1] and set $\tau_{fail}=\lceil D / q \rceil = D$ (c.f. Appendix D in [R1]).
> > > We want to highlight that applying such a trigger for time-varying functions has not been published before and therefore this algorithm is an additional contribution and not a baseline.
> > > We benchmarked this trigger for various rates of change as well as different dimensions.
> > > The results are in Appendix A.3 (Figure 9).
> > > We can observe that the performance-based event trigger struggles in the low-dimensional setting.
> > > Also in the high-dimensional setting, it is only able to match the performance of R-GP-UCB.
> > > We believe that further adding heuristics, such as including the current noise level in determining $\tau_{fail}$, can improve the performance of this event trigger.
> > > Therefore, we remain optimistic that this is a research direction worth pursuing. The intriguing part is the decoupled nature of the event trigger from the GP prior.
> > > In this simple current form, however, this comes at the cost of high regret.
> > >
> > > (regarding the last point see next comment)

---

> ### Author Response · Authors · 2024-11-21
>
> (for the first two points see previous comment)
>
> > Details of Figure 11. Figure 11 still lacks experimental details, such as which benchmarks/datasets are used in Figure 11. Additionally, could the authors provide an intuitive explanation of why hyperparameter settings have such a minimal impact on performance?
>
> For the sensitivity analysis, we used within-model objective functions to solely focus on the effect of the hyperparameter $\delta_B$.
> This detail has now been added to the text, as we overlooked it during the last revision.
> All hyperparameters are listed in Table 12. Also note that the figure is now Figure 13 given the above mentioned additional results.
> Our intuition and interpretation are the following.
> We attribute the small influence on the performance to the fact that while the average number of resets increases (as it should when loosening the high-probability bound), this increase remains modest for lower $\varepsilon$.
> Hence, the impact on regret is also small.
> The increase in average number of resets is more noticeable for $\varepsilon = 0.05$.
> However, given the result from Figure 5 (reset frequency vs. reset timing), we expect that especially at higher rates of change, the "correct" reset timing (given the respective $\delta_B$) is the main factor for reduced regret compared to e.g. R-GP-UCB.
>
> ---
> # References
>
> [R1] Eriksson, David, et al. "Scalable global optimization via local Bayesian optimization." Advances in neural information processing systems 32 (2019).

---

> > ### Comment · Reviewer_TDqt · 2024-11-27
> > **Reply to the changes**
> >
> > Thanks for the response and the effort you’ve put in. Here are my suggestions:
> >
> > * The hyperparameter setting of a TuRBO-like event trigger is not suitable. For a 2D problem, set $\tau_{fail}=D=2$ is obviously too small, leading to too many unnecessary resets. Note TuRBO uses $\tau_{fail}=D$ to shrink the trust region rather than reset the entire dataset. In fact, it resets only after $\tau_{fail}$ is triggered 7 times. So in your case, set $\tau_{fail}=D*k+b$ is more reasonable where $k$ and $b$ are hyperparameters, e.g. k can be set in the range of 1-7.
> >
> > * Using within-model objective functions for the sensitivity analysis is not enough. As the proposed algorithm relies on prior GP assumptions, it is meaningful to test whether a fixed hyperparameter setting is robust enough for real applications which might be out of the assumptions.

---

> > > ### Author Response · Authors · 2024-11-28
> > >
> > > We thank the reviewer for their comments and the continued discussion.
> > >
> > > ---
> > >
> > > > On the TuRBO-like event trigger
> > >
> > > We agree that this setting led to too many resets and thus bad performance for a 2D problem. This was a misunderstanding, as we thought the reviewer explicitly requested that the main mechanism of TuRBO, i.e., changing the trust region performance-based, should be used for designing the performance-based event trigger.
> > > The new proposed parametrization of the performance-based trigger introduces two additional hyperparameters that have to be tuned for each problem and it seems likely these parameters have a large influence on the performance i.e. a high sensitivity regarding the choice of $k$ and $b$.
> > > We believe that tuning these hyperparameters for good empirical performance is beyond the scope of this paper given that the performance-based trigger is not an existing baseline but would be a contribution on its own.
> > > We remain positive about the prospect of such performance-based triggers for the mentioned reasons but designing them in a principled way as well as benchmarking them sufficiently is a new and different contribution from what we propose in our paper.
> > >
> > > ---
> > >
> > > > On the sensitivity of $\delta_B$ and our sensitivity analysis
> > >
> > > Our reasoning behind using within-model functions was that we explicitly consider Assumption 2 as our regularity assumption on the temporal changes--also for our theoretical results. Therefore, it seemed also fitting to consider this setting for the sensitivity analysis of our event trigger. This isolates the effects of the hyperparameter from other design choices such as kernel choices and hyperpriors.
> > >
> > > Generally, we demonstrate the robustness of the hyperparameter by using the same parameter ($\delta_B=0.1$) _for all experiments_, including the application examples in the paper, which included the temperature data benchmark, a policy search benchmark of a cart-pole, and a stock market benchmark from prior work.
> > > Although we did not tune the hyperparameter for any of the experiments, the algorithms still performed well.
> > > We also did not change this parameter for the newly requested benchmarks (Ackley and Hartmann), and the algorithms show good performance here.
> > > Since we never change the hyperparameter and our proposed method still exhibits good performance this does demonstrate a certain robustness of the hyperparameter also for settings beyond the regularity assumptions in Assumption 2.

---

### Review · Reviewer_P7Dp · 2024-11-13

**Summary Of Contributions:**

This paper considers the problem of time-varying Bayesian optimization (TVBO), where at each step $t$, a learner queries an oracle with a point $x_t$ and is given back a noisy observation $y_t$ of the function value $f_t (x_t)$, and the aim is to minimize the dynamic regret $\min_{x_1, x_2, ...} \sum_{t=1}^{T} [\max_{x} f_t (x) - f_t (x_t)]$. The functions $f_t$ come from a smoothly-varying Gaussian process, as defined in Assumption 2, where a parameter $\epsilon$ limits how much the functions $f_t$ differ from one step to another. The observations $y_t$ are corrupted with Gaussian noise (Assumption 1). The functions also satisfy some standard smoothness assumptions (Assumption 3). The main issue this paper considers is how to solve this problem without explicit knowledge of $\epsilon$, here "solve" generally means obtain a regret rate that is worst-case bounded as $T \phi(\epsilon)$ for some function $\phi$ of $\epsilon$ such that $\phi(\epsilon) \to 0$ when $\epsilon \to 0$. This is because in general, linear regret is unavoidable in this problem. In order to remove the knowledge of $\epsilon$, the authors develop a new algorithm (Event-triggered GP-UCB, Algorithm 1) that adaptively resets the stored dataset whenever the data has become too stale, according to a certain threshold test. The paper also conducts extensive experimental evaluations for the algorithm.

**Audience:**

Yes

**Broader Impact Concerns:**

N/A.

**Claims And Evidence:**

Yes

**Requested Changes:**

Please add more motivation and address the questions I mentioned in the previous section, if possible. This is not critical, I think the world is strong enough to recommend it for acceptance as-is.

**Strengths And Weaknesses:**

1. The algorithm is well motivated, and the problem is important. In most cases, parameters like $\epsilon$ are rarely known ahead of time and removing them is always useful.
2. ET-GP-UCB shows low sensitivity to the main parameter, $\delta_{B}$, the threshold at which resets happen. The experiments also show ET-GP-UCB displays large robustness to $\epsilon$ changes whereas previous algorithms (that requires precise knowledge of $\epsilon$ do not.

Questions:
1. How do we know $r$, $a_1$, and $b_1$ in Theorem 1?
2. Why are we considering only the dynamic regret? Can we achieve sublinear regret if we allow for regret that tracks comparators which are fixed for certain windows?
3. How do we know the upper and lower bounds on $\epsilon$? Are they available for practical problems?
4. We assume that $\sigma_n$ is known in (11), but how do we know it?

Minor points
1. The $t^\prime$ notation in p. 6 is a bit confusing, it might be better to be more explicit and name the reset timesteps at $t_1, t_2, \ldots$ and consider $t \in [t_i, t_{i+1}]$.
2. The section on the empirical impacts of $\underline{N}$ and $\overline{N}$ on the reset times (p. 7) might be better moved to the experimental evaluation section.
3. In Appendix D, there is a missing reference "As stated in Sec. ??".

---

> ### Author Response · Authors · 2024-11-15
>
> Thank you for taking the time and thouroughly reviewing our paper. In the following, we aim to answer all questions as well as address your comments.
>
> > 1. How do we know $r$, $a_1$, $b_1$ in Theorem 1?
>
> $r$ is the length of the hypercube $\mathbb{X}$, which we set in our within-model comparisons to $1$ to consider the unit hypercube.
> This is without loss of generality as one can always rescale the input space $\mathbb{X}$ to such a unit hypercube. This is recommended in practical BO implementations to normalize the data [see BOTorch].
>
> The parameters $a_1$ and $b_1$ are kernel dependent and exist for kernels that are at least four times (c.f. Theorem 5 in S. Ghosal and A. Roy 2006 [R1]).
> It is also possible to consider other high-probability bounds on the Lipschitz constants, e.g., those used in Lederer et al. 2019 [R2]. Here, one would need to restate Lemma 6 accordingly.
>
>
> > 2. Why are we considering only the dynamic regret? Can we achieve sublinear regret if we allow for regret that tracks comparators that are fixed for certain windows?
>
> Great question. We have considered dynamic regret as a first step toward regret bounds for algorithms without prior knowledge of epsilon. Extending this to multiple comparators that are fixed for a certain window is very interesting but something we have not explored yet. Such a setting should be especially interesting for adaptive approaches as proposed herein. Intuitively, given regularity assumptions on the length of the window in relation to the problem dimension as well as underlying length scales, sublinear regret should be possible. Such regularity assumptions should give an algorithm enough time to match the performance of the current comparator; hence, sublinear regret could be possible. Again, we have yet to explore this direction, but we consider it very interesting.
>
> > 3. How do we know the upper and lower bounds on epsilon? Are they available for practical problems?
>
> Choosing tight upper and lower bounds on epsilon can be challenging for practical problems. The benefit of using our event trigger is that, empirically, we do not need tight bounds on $\epsilon$ since the performance is not sensitive to them (see Table 1). We achieve the best empirical performance without any bounds! We attribute this to the theoretically grounded design of our event trigger.
> So, while the theoretical regret bounds still benefit from some knowledge about $\epsilon$, the proposed event trigger, in practice, does not rely on them for good empirical performance.
>
> > 4. We assume that the noise is known in (11), but how do we know it?
>
> Ideally, the noise can be estimated from some hold-out data. This is how we obtained the noise value for the temperature example. If this is not possible, the noise value has to be estimated online. There are a few different ways how to estimate the noise, which we further discuss in Sec. 6.5. The two main ideas are to use reasonable hyperpriors on the noise and/or use a learn-then-monitor approach for learning the noise value online. Another idea that we, however, have not explored is to consider a Student-t process instead of a Gaussian process as the surrogate model. Student-t processes are known to perform well in settings with unknown noise. Here, one could similarly leverage uniform error bounds bounds on the posterior to design an event trigger. Obtaining regret bounds for this surrogate model is however likely more involved.
>
> # Minor points
>
> > 1. The notation in p. 6 (for the time after the last reset) is a bit confusing, it might be better to be more explicit.
>
> Thank you for the suggestion. We hope this is more clear in the revised version.
>
> > 2. The section on the empirical impacts of the upper and lower bounds on the reset times (p. 7) might be better moved to the experimental evaluation section.
>
> This is a valid point. We have moved them to the experimental section under the subsection "Empirical Impact of \bar N and \ubar N on Reset Times and Regret"
>
> > 3. In Appendix D, there is a missing reference "As stated in Sec. ??".
>
> Thank you for spotting this! We have fixed the reference in the revised version.
>
> ---
> # References
>
> [R1] Ghosal, S., & Roy, A. (2006). Posterior consistency of Gaussian process prior for nonparametric binary regression. The Annals of Statistics, Vol. 34, No. 5, 2413–2429
>
> [R2] Lederer, A., Umlauft, J., & Hirche, S. (2019). Uniform error bounds for Gaussian process regression with application to safe control. Advances in Neural Information Processing Systems, 32.

---

### Review · Reviewer_ANPd · 2024-11-14

**Summary Of Contributions:**

The paper proposes an algorithm for non-stationary kernelized bandits. The algorithm uses a test to detect if the reward function has changed substantially. The paper shows sub-linear regret bounds for the proposed algorithm.

**Audience:**

Yes

**Claims And Evidence:**

No

**Requested Changes:**

1. The abstract and introduction need to be updated to accurately state the quality of the performance bounds (not fully adaptive to the rate of change).

2. How would your performance bounds look like in the multi-armed bandit case?

3. Authors need to discuss the form of the Event-Trigger condition and why it can use only one datapoint.

**Strengths And Weaknesses:**

1. The abstract and introduction promise regret bounds that are adaptive to rate of change. But Theorem 1 and Corollary 1 require upper and lower bounds on the rate of change. The following papers show (truly) adaptive regret bounds for non-stationary multi-armed bandits (only one is cited):

Peter Auer, Pratik Gajane, Ronald Ortner. Adaptively tracking the best bandit arm with an unknown number of distribution changes. COLT, 2019
Yifang Chen, Chung-Wei Lee, Haipeng Luo, Chen-Yu Wei. A new algorithm for nonstationary contextual bandits: Efficient, optimal and parameter-free. COLT, 2019.
Chen-Yu Wei and Haipeng Luo. Non-stationary reinforcement learning without prior knowledge: an optimal black-box approach. COLT, 2021.
Joe Suk and Samory Kpotufe. Tracking most significant arm switches in bandits. COLT, 2022.
Yasin Abbasi Yadkori, Andras Gyorgy, Nevena Lazi. A New Look at Dynamic Regret for Non-Stationary Stochastic Bandits. JMLR, 2023.

2. The Event-Trigger condition in Algorithm 1 seems to consider only one datapoint. Is this correct? Perhaps in this work, and given the Gaussian noise condition, such a condition might be enough to obtain sub-linear regret bounds. But more generally (e.g. Bernoulli observations) such a condition does not lead to sub-linear regret bounds. The above papers can show strong adaptive results only because they consider collection of observations over time intervals.

---

> ### Author Response · Authors · 2024-11-15
>
> Thank you for reading our paper and for your feedback. In the following, we hope to address your questions and clarify key aspects of our results.
>
> The algorithm that we propose does _not_ yield sub-linear regret. As shown in prior work sublinear regret is unattainable under Assumption 2. The known lower bound on the expected cumulative regret is $\Omega(T\varepsilon)$. We discuss this lower bound in more detail in the second paragraph of our introduction. Because the algorithm queries the objective at only one point per time step and the function also varies at each time step, convergence to the exact optimum is not feasible, resulting in an unavoidable linear gap in simple regret.
> Our problem setting is more similar to the restless MAB setting, which is known to be intractable [R1,R2] without further regularity assumptions.
>
> ## Requested Changes
>
> > 1. The abstract and introduction need to be updated to accurately state the quality of the performance bounds (not fully adaptive to the rate of change).
>
> Thank you for pointing this out. To avoid misunderstanding: Our algorithm is adaptive to the rate of change and does not rely on the exact prior knowledge of this rate (see Table 1). The regret bounds are also dependent on the rate of change and are for the more general case of any strategy that resets in a given window where this window depends on some prior knowledge about bounds for the rate of change. We make this precise in the introduction and abstract as well as Section 4 (specifically the sentences preceding Theorem 1). We adjusted the abstract slightly to make this clearer. Additionally, we have now clarified this point further in the introduction (preceding the statement of contributions on page 2) by explicitly stating that we require lower and upper bounds on the rate of change.
>
> > 2. How would your performance bounds look like in the multi-armed bandit case?
>
> In our paper, we focus on the problem of spatially correlated arms through a Gaussian process motivated by time-varying problems with an infinite number of arms, such as a policy search in a time-varying setting where a feedback policy is parameterized by values in R^d.
>
> However, we recognize the conceptual relevance of dynamic MABs and, in response, have added a discussion on this similarity (and the differences) in the related work section.
> For example, people have studied the above-mentioned restless bandits, for which no sublinear regret is possible. This is similar to our setting of ongoing changes. Therefore, adaptive algorithms similar to ours (but tailored to the MAB setting) will still yield inevitable linear regret.
> For this new discussion the suggested references were very helpful and are now included.
>
> > 3. Authors need to discuss the form of the Event-Trigger condition and why it can use only one datapoint.
>
> For our event trigger it is sufficient to only consider the current data point in the event trigger condition because the GP (high-probability) error bounds are uniform for all $t$ and $x$ [R3]. We interpret the cause for a violation of these bounds as a change. Naturally, there will be false positives and false negatives.
>
> On an intuitive level, we aim to test if the new point is consistent with the Gaussian process posterior, which is conditioned on all past data points.
> If this is not the case, we conclude that there has been a change with high probability. Empirically, this works quite well, as we show in Section 6. In Section 6.3, we extend ET-GP-UCB to retain more information. In some sense, this strategy does look at more data points and, after the violation of the bounds, retroactively decides where the change happened.
>
> To increase clarity on this point we added a small discussion after the design of the event trigger in Section 5 to clarify this. We further motivate the choice of trigger at the beginning of Section 5.
> Given the framework provided by Definition 2, it is possible to include past points in the event trigger. One could, e.g., evaluate the trigger bound for all past data points simultaneously. However, given that the GP is fitted on these past data points, an evaluation of the event trigger condition to be true is unlikely as notable changes would generally have been detected earlier. Still, it is possible to include more data points--just not from the current time step as the algorithm can only query the objective once.
>
> ---
> # References
>
> [R1] Whittle, Peter. "Restless bandits: Activity allocation in a changing world." Journal of Applied Probability 25.A (1988): 287-298.
>
> [R2] Slivkins, Aleksandrs, and Eli Upfal. "Adapting to a Changing Environment: the Brownian Restless Bandits." COLT. 2008.
>
> [R3] Srinivas, Niranjan, et al. "Gaussian Process Optimization in the Bandit Setting: No Regret and Experimental Design." Proceedings of the 27th International Conference on Machine Learning. 2010.

---

> > ### Comment · Reviewer_ANPd · 2024-11-20
> > **Response**
> >
> > Thanks for your response.
> >
> > * Sublinear regret: if \epsilon is a constant, then of course regret is linear in time. But the interesting regime is when \epsilon=o(1). In literature on K-armed non-stationary bandits, we have the following results: regret is O(\sqrt{KST}) when we have S changes in the identity of the optimal arm, and regret is O((KV)^{1/3} T^{2/3} ) when the total variation in the reward function is V. Obviously, if S and V scale as O(T), regret is linear. Your problem is closely related to obtaining dynamic regret in terms of total variation.
> >
> > * The case of multi-armed bandits: I understand that your main motivation is the Gaussian process bandits. But the multi-armed case is just a special case. No? I would like to see your regret bounds for this special case, and how it compares to the above total variation bounds.
> >
> > * The Event-Trigger condition: again, your problem is closely related to obtaining dynamic regret in terms of total variation. To have a near-optimal performance, an algorithm needs to consider collection of observations over time intervals, without which near-optimal performance is not possible.

---

> > > ### Author Response · Authors · 2024-11-21
> > >
> > > Thank you for your reply and engaging in the discussion.
> > >
> > > > Sublinear regret: if \epsilon is a constant, then of course regret is linear in time. But the interesting regime is when \epsilon=o(1). In literature on K-armed non-stationary bandits, we have the following results: regret is O(\sqrt{KST}) when we have S changes in the identity of the optimal arm, and regret is O((KV)^{1/3} T^{2/3} ) when the total variation in the reward function is V. Obviously, if S and V scale as O(T), regret is linear. Your problem is closely related to obtaining dynamic regret in terms of total variation.
> > >
> > > The setting of $\epsilon = o(1)$ is more similar to the setting of a finite variational budget which we discuss in the introduction, related work, and Appendix A.5. In Appendix A.5, we also present empirical comparisons with GP-UCB type algorithms for this setting, demonstrating that ET-GP-UCB significantly outperforms them. This highlights the potential of event-triggered BO under fixed variational budgets and motivates further research in this direction.
> > >
> > > However, despite the fact that sublinear regret is unavoidable for constant $\epsilon$, we still find this case compelling due to its relevance to many real-world continual learning problems, which we believe are particularly intriguing. For this reason, our paper explicitly focuses on this setting. Furthermore, we are not the first to study this scenario, as we elaborate in the second paragraph of the related work section.
> > >
> > > > The case of multi-armed bandits: I understand that your main motivation is the Gaussian process bandits. But the multi-armed case is just a special case. No? I would like to see your regret bounds for this special case, and how it compares to the above total variation bounds.
> > >
> > > To some extent, MABs can indeed be viewed as a special case of GP bandits with no correlation between the arms. However, classic MABs are, in a sense, an edge case where the standard proof techniques for GP bandits, which leverage GP-specific uniform error bounds, do not apply effectively. In the MAB setting, the lack of correlations means that the kernel-specific bound on information gain $\gamma$ (see Equation (34) or Section 5 in [R3]) grows linearly with $T$. Figure 3 in [R3] illustrates how different kernels influence the bound on the maximum information gain. The sublinear trend for various kernels (eg. for those from the Matern class) is the key factor for sublinear regret in algorithms such as GP-UCB.
> > >
> > > Because of the linear bound on the mutual information gain, the regret bounds in our Theorem 1 become vacuous in the special case of independent arms, as they rely on correlations modeled by the kernel. Analyzing MABs requires fundamentally different proof techniques tailored to this setting.
> > >
> > > While we acknowledge the importance of MABs, the focus of our paper is on GP bandits. Thus, we consider deriving MAB-specific regret bounds for adaptive resets beyond the scope of this work.
> > >
> > > > The Event-Trigger condition: again, your problem is closely related to obtaining dynamic regret in terms of total variation. To have a near-optimal performance, an algorithm needs to consider collection of observations over time intervals, without which near-optimal performance is not possible.
> > >
> > > We acknowledge the connection and appreciate the insights. As noted, we do not consider the case where $\epsilon = o(1)$. Additionally, we recognize that our regret bounds are not optimal for any specific event-trigger condition. Instead, they represent a worst-case analysis applicable to any possible trigger condition. This limitation is explicitly discussed following Corollary 1 in our paper.
> > >
> > > [R3] Srinivas, Niranjan, et al. "Gaussian Process Optimization in the Bandit Setting: No Regret and Experimental Design." Proceedings of the 27th International Conference on Machine Learning. 2010.

---

> > > > ### Comment · Reviewer_ANPd · 2024-12-15
> > > >
> > > > It looks like authors don't want to answer the following questions: What is your total variation regret? (this is concerned with \epsilon=o(1) regime) What is the regret bounds for the special case of multi-armed bandits?
> > > >
> > > > Given the form of your Event-Trigger condition, I expect your rates to be very sub-optimal.

---

> > > > > ### Author Response · Authors · 2024-12-16
> > > > >
> > > > > > Regarding the first point
> > > > >
> > > > > The variational budget setting (or total variation) is typically studied under different regularity assumptions. Previous work has often adopted a frequentist or agnostic perspective, where formulating the variational budget can be easily done (as discussed on page 4 of our paper).
> > > > >
> > > > > In our setting, one could allow $\epsilon$ to decay over time, for instance, by setting $\epsilon = 1 / t^2$ in (37), and study the resulting behavior.
> > > > > However, we do not explore this direction in the paper, as our focus is on continuous, ongoing changes. It is worth noting that if $\epsilon$ were to change over time, the upper and lower bounds on the resets would have to be adjusted over time accordingly, as implied by Corollary 1.
> > > > >
> > > > > In general, in our paper, we adopt a Bayesian perspective, assuming that f is a sample from a spatio-temporal Gaussian Process (GP), where the temporal kernel follows Assumption 2 with a fixed $\epsilon$. This assumption is also considered in various other works (see end of page 3, top of page 4)--also without a consideration of the variational budget setting.
> > > > >
> > > > > We believe both settings to be relevant. From a theoretical point, the fixed variational budget setting may be more appealing as sublinear regret can be proven; however, if the changes are ongoing, such algorithms would likely degrade in performance over time: if the variational budget is exhausted the resulting algorithms start to only exploit and not adjust to the changes.
> > > > > Empirically, we outperform such approaches in Section A.5.
> > > > >
> > > > > > Regarding the second point
> > > > >
> > > > > Adapting GP-UCB-like bounds to the MAB setting is unfortunately not meaningful.
> > > > > Even translating classic results would yield trivial regret bounds as a main object in these GP-UCB-like bounds is the mutual information gain which for MABs is zero. Hence, the maximum information gain does not decrease over time but grows linearly. The resulting bound will be trivially linear. For example, see [R3, Theorem 1], which is a classic GP-UCB bound for a finite but correlated action space: if $\gamma_T = O(T)$, then the bound becomes linear.
> > > > > The same arguments hold for our bound: If $\gamma_{\bar N} = O(\bar N)$, the dependence on $\epsilon$ would cancel out and we obtain a trivial linear bound (see e.g. (40)).
> > > > >
> > > > > ---
> > > > >
> > > > > Lastly, we want to clearly state that in this paper, we do not study the variational budget setting as this significantly differs from the problem setting that we consider and for which we develop the proposed algorithm.
> > > > > We will also not include additional regret bounds for MABs. We believe both would significantly defer from the intent of our paper, i.e., a practical algorithm for time-varying BO for continuous changes that is based on theoretical insights and yields good empirical performance on various problems.

---

### Author Response · Authors · 2024-11-15

# General Comments

We thank all reviewers for their time and effort to review our paper.
In this general comment, we summarize the main changes made in the revised version and aim to address all specific points in the individual rebuttals.
To better parse the new version, we marked all content-related changes in red. The main changes include:

- A new experiment with a change in the rate of change in Appendix A.2 which further demonstrates the adaptive nature of our approach.
- We added performance plots over time for the ablation study of the event trigger hyperparameter $\delta_B$ to show how it affects the optimization process in Appendix B.
- In Section 3, we added a discussion on dynamic and nonstationary multi-armed bandits, highlighting similarities and differences in problem settings.
- We added a comparison to a performance-based event trigger in Appendix A.3.
- We ran additional experiments on standard benchmarks, namely Ackley and Hartmann, and show the results in Appendix A.4.

---

### Decision · Action_Editor_qSTq · 2024-12-20

**Recommendation:** Accept with minor revision

**Comment:**

The paper describes an event-triggered algorithm to detect shift in the objective function for time-varying Bayesian optimization approaches.
During the review phase, reviewers raised several concerns about the paper:

Reviewer ANPd expressed doubts regarding whether the presented results could be directly translated to the multi-armed bandit (MAB) scenario, which can be viewed as a special case of the Gaussian process (GP)-based Bayesian optimization setting considered in the paper. I support the authors' argument that, while the two scenarios appear similar, translating the regret bounds derived from GP-UCB is non-trivial and beyond the scope of this work. The revised version of the paper also includes a discussion relating to MAB and highlights the key differences.

Reviewer TDqt pointed out that performance-based reset strategies are a practical tool for restarting optimization. In response, the authors included a comparison with a simple strategy inspired by prior work. Reviewer TDqt raised concerns about the evaluation of this baseline and proposed a more sophisticated approach. While I agree that the baseline could be further optimized, I believe this refinement would not fundamentally alter the main contributions of the paper, which center on providing a theoretical framework for detecting shifts in the objective function.

Secondly, Reviewer TDqt argued that the sensitivity analysis of hyperparameters was limited. However, my understanding is that this concern primarily pertains to the general modeling assumptions inherent in Gaussian processes, rather than specific assumptions made by the proposed methods. While I acknowledge the validity of these concerns, they have already been extensively addressed in the existing literature.

As discussed above, I do not believe that the paper overstates its claims and find that it provides both theoretical and empirical evidence to substantiate them. However, to minimize potential confusion for future readers, I recommend that the authors:

 -   Rephrase the abstract, which currently states that the proposed method outperforms state-of-the-art methods. Since the comparison is limited to GP-UCB, which is not considered the current state-of-the-art in Bayesian optimization, this claim should be adjusted.
 -  Add a sentence to Section A.3 noting that performance-based triggers could be further optimized, potentially leading to improved performance.

**Audience:**

Bayesian optimization is an active area of research in the field of machine learning and hence I am convinced that some individuals of  TMLR's audience will be interested in the paper

**Claims And Evidence:**

The paper provides provides both theoretical and empirical evidence to substantiate the claims made. See more detailed discussion below.